# Sepsis leads to lasting changes in phenotype and function of memory CD8 T cells

Isaac J Jensen[1], Xiang Li[2], Patrick W McGonagill[3], Qiang Shan[4], Micaela G Fosdick[5], Mikaela M Tremblay[6], Jon CD Houtman[5,6], Hai-Hui Xue[4], Thomas S Griffith[7,8,9,10,11], Weiqun Peng[2], Vladimir P Badovinac[1,6]*

[1]Department of Pathology, University of Iowa, Iowa City, United States; [2]Department of Physics, The George Washington University, Washington, United States; [3]Department of Surgery, University of Iowa, Iowa City, United States; [4]Center for Discovery and Innovation, Hackensack University Medical Center, Nutley, United States; [5]Interdisciplinary Graduate Program in Molecular Medicine, University of Iowa, Iowa City, United States; [6]Interdisciplinary Graduate Program in Molecular Medicine, University of Iowa, Iowa City, United States; [7]Microbiology, Immunology, and Cancer Biology PhD Program, University of Minnesota, Minneapolis, United States; [8]Department of Urology, University of Minnesota, Minneapolis, United States; [9]Center for Immunology, University of Minnesota, Minneapolis, United States; [10]Masonic Cancer Center, University of Minnesota, Minneapolis, United States; [11]Minneapolis VA Health Care System, Minneapolis, United States

*For correspondence: vladimir-badovinac@uiowa.edu

Competing interest: The authors declare that no competing interests exist.

**Abstract** The global health burden due to sepsis and the associated cytokine storm is substantial. While early intervention has improved survival during the cytokine storm, those that survive can enter a state of chronic immunoparalysis defined by transient lymphopenia and functional deficits of surviving cells. Memory CD8 T cells provide rapid cytolysis and cytokine production following re-encounter with their cognate antigen to promote long-term immunity, and CD8 T cell impairment due to sepsis can pre-dispose individuals to re-infection. While the acute influence of sepsis on memory CD8 T cells has been characterized, if and to what extent pre-existing memory CD8 T cells recover remains unknown. Here, we observed that central memory CD8 T cells ($T_{CM}$) from septic patients proliferate more than those from healthy individuals. Utilizing LCMV immune mice and a CLP model to induce sepsis, we demonstrated that $T_{CM}$ proliferation is associated with numerical recovery of pathogen-specific memory CD8 T cells following sepsis-induced lymphopenia. This increased proliferation leads to changes in composition of memory CD8 T cell compartment and altered tissue localization. Further, memory CD8 T cells from sepsis survivors have an altered transcriptional profile and chromatin accessibility indicating long-lasting T cell intrinsic changes. The sepsis-induced changes in the composition of the memory CD8 T cell pool and transcriptional landscape culminated in altered T cell function and reduced capacity to control *L. monocytogenes* infection. Thus, sepsis leads to long-term alterations in memory CD8 T cell phenotype, protective function and localization potentially changing host capacity to respond to re-infection.

## Introduction

Dysregulated systemic inflammatory responses define septic events and the associated cytokine storm, which is comprised of both pro- and anti-inflammatory cytokines (*CDC, 2020*; *Singer et al., 2016*). Sepsis leads to a substantial global health and economic burden wherein nine people develop

**eLife digest** A dirty cut, a nasty burn, a severe COVID infection; there are many ways for someone to develop sepsis. This life-threatening condition emerges when the immune system overreacts to a threat and ends up damaging the body.

Even when patients survive, they are often left with a partially impaired immune system that cannot adequately protect against microbes and cancer; this is known as immunoparalysis. Memory CD8 T cells, a type of immune cell that is compromised by sepsis, are a long-lived population of cells that 'remember' previous infection or vaccination, and then react faster to prevent the same illness if the person ever encounters the same threat again. Yet it is unclear how exactly sepsis harms the function and representation of memory CD8 T cells, and the immune system in general.

Jensen et al. investigated this question, first by showing that sepsis leads to a profound loss of memory CD8 T cells, but that surviving memory CD8 T cells multiply quickly – especially a subpopulation known as central memory CD8 T cells – to re-establish the memory CD8 T cell population. Since the central memory CD8 T cells proliferate better than the other memory T cells this alters the overall composition of the pool of memory CD8 T cells, with central memory cells becoming overrepresented.

Further experiments revealed that this biasing toward central memory T cells, due to sepsis, created long-term changes in the distribution of memory CD8 T cells throughout the body. The way the genetic information of these cells was packaged had also been altered, as well as which genes were switched on or off. Overall, these changes reduced the ability of memory CD8 T cells to control infections.

Together, these findings help to understand how immunoparalysis can emerge after sepsis, and what could be done to correct it. These findings could also be applied to other conditions – such as COVID-19 – which may cause similar long-term changes to the immune system.

sepsis every 6 s and two of those individuals die (*Rudd et al., 2020*). In the United States alone the cost to treat sepsis is >$20 billion with a mortality rate of ~20 % (*CDC, 2020*). While a 20 % mortality rate is high, it is also a vast improvement over the last 30 years where mortality had been at ~50 % (*Dombrovskiy et al., 2007*; *Gaieski et al., 2013*). This reduction in mortality rate has largely been through early intervention as the complexity of the cytokine storm has, dishearteningly, lead to the failure of >100 phase II and III clinical trials targeting the pro-inflammatory aspects of the cytokine storm (*Marshall, 2014*). Yet, even as survival of the cytokine storm has increased it has also become apparent that previously septic individuals are still at increased risk for mortality, this defines the sepsis-induced immunoparalysis state (*Delano and Ward, 2016a*; *Delano and Ward, 2016b*; *Dombrovskiy et al., 2007*; *Donnelly et al., 2015*).

Sepsis-induced immunoparalysis is characterized by an increased susceptibility to both new and previously encountered unrelated infections and cancer (*Danahy et al., 2019*; *Jensen et al., 2018a*; *Kutza et al., 1998*; *Walton et al., 2014*). Alternately, sepsis-induced immunoparalysis reduces susceptibility to development of autoimmunity, cumulatively demonstrating immunologic impairment (*Jensen et al., 2020*). These profound impairments are sufficient to reduce the 5 year survival of septic cohorts, relative to non-septic cohorts; consequently, the majority of sepsis-associated mortality is late mortality secondary to the cytokine storm (*Dombrovskiy et al., 2007*; *Donnelly et al., 2015*; *Gaieski et al., 2013*). This immunologic impairment is typified by transient lymphopenia and reduced capacity of various surviving lymphocyte populations to perform effector function (*Hotchkiss et al., 2016*; *Hotchkiss et al., 2013*), including CD4 (*Cabrera-Perez et al., 2014*; *Cabrera-Perez et al., 2015*; *Chen et al., 2017*; *Jensen et al., 2020*; *Martin et al., 2020*; *Sjaastad et al., 2020b*) and CD8 T cells (*Condotta et al., 2013*; *Danahy et al., 2017*; *Danahy et al., 2019*; *Duong et al., 2014*; *Serbanescu et al., 2016*; *Xie et al., 2019*), B cells (*Hotchkiss et al., 2001*; *Sjaastad et al., 2018*; *Unsinger et al., 2010*), NK cells (*Hou et al., 2014*; *Jensen et al., 2021b*; *Jensen et al., 2018b*; *Souza-Fonseca-Guimaraes et al., 2012*), and dendritic cells (DCs) (*Poehlmann et al., 2009*; *Roquilly et al., 2017*; *Strother et al., 2016*). We and others have characterized numerous impairments early after sepsis induction; however, the extent to which those cell populations recover in number and function remains largely unknown.

Specifically, sepsis-induced lymphopenia impacts both memory and naïve CD8 T cells early after sepsis (*Condotta et al., 2015*; *Condotta et al., 2013*; *Duong et al., 2014*; *Jensen et al., 2018a*; *Markwart et al., 2014*). Additionally, those memory CD8 T cells that survive the cytokine storm are less capable of undergoing antigen-dependent effector function and responding to inflammatory cues (bystander activation). These intrinsic impairments, in conjunction with the numeric deficits imposed by the lymphopenic environment, reduce host capacity to control both infection (i.e. viral and bacterial) and cancer (*Danahy et al., 2017*; *Danahy et al., 2019*; *Duong et al., 2014*; *Gurung et al., 2011*). Additionally, extrinsic factors, such as reduced integrin expression on endothelia (*Danahy et al., 2017*) or altered monocyte/ macrophage activity (*Jensen et al., 2021a*; *Roquilly et al., 2020*), can influence CD8 T cell capacity to migrate into sites of infection. Even when T cells are spared from the cytokine storm by vascular exclusion (i.e. tissue residence) CD8 T cell-mediated protection can be hampered by inability of other cells (e.g. endothelia) to respond to the inflammatory cues provided by CD8 T cells (*Danahy et al., 2017*). Yet, these impairments are largely characterized proximal to the septic insult. However, sepsis-induced impairments are long-lasting and may not be consistent across time (*Jensen et al., 2018a*). Specifically, the lymphopenic environment is transient yet the ability to control cancer can remain reduced long after numeric recovery is complete (*Danahy et al., 2019*). Thus, while there does not appear to be preferential susceptibility to sepsis, if and how different subsets of memory CD8 T cells recover may dramatically shape how hosts respond to pathogen re-encounter and thereby contribute to the immunoparalysis state.

Here, using samples from septic patients and well described experimental models we demonstrate increased proliferation of CD8 T cells (particularly central memory cells [$T_{CM}$]) in septic patients and mice after cecal ligation and puncture (CLP)-induced sepsis, relative to non-septic controls. As a consequence of this increased proliferation, there is a remodeling of the memory CD8 T cell pool. This compositional change in turn leads to lasting changes in the localization, function, and protective capacity of pre-existing memory CD8 T cells.

## Results

### Increased CD8 T cell proliferation in septic patients

The sepsis-induced immunoparalysis state poses a substantial threat to the health and long-term survival of septic patients (*Delano and Ward, 2016a*; *Delano and Ward, 2016b*; *Dombrovskiy et al., 2007*; *Donnelly et al., 2015*). A major contributing factor to sepsis-induced immunoparalysis is the intrinsic and numerical deficits imposed on naive and memory CD8 T cells (*Jensen et al., 2018a*). In particular, deficits in existing memory CD8 T cells can enhance host susceptibility to pathogens against which the host was previously immune or vaccinated. To understand how CD8 T cells respond to septic insult and the lymphopenic state, septic patients were recruited within 24 hr of admission and the frequency and number of CD8 T cells in the peripheral blood were compared to that of healthy controls. Patient cohorts did not exhibit substantial demographic differences though septic patients were severely ill, as defined by APACHE II and SOFA scores (*Table 1*). While there was not a difference in the frequency of CD8 T cells among lymphocytes between septic patients and healthy controls (*Figure 1a and b*), there was a cohort of septic patients with a substantially reduced number of CD8 T cells per mL of blood (*Figure 1c*) reflecting the sepsis-induced lymphopenia. It is relevant to consider that admission time may not correspond to the onset of sepsis such that admitted patients may have not yet experienced or already recovered from sepsis-associated lymphopenia. Thus, numeric variability in samples may reflect a broad range of insult and recovery within the 24 hr of admission. Notably, robust induction of Ki67 expression, a marker of recent proliferation, by CD8 T cells (*Figure 1a and d*) was observed, regardless of degree of lymphopenia.

**Table 1.** Patient demographics.

| Patients | Septic (n = 27) | Control (n = 16) | p-value |
|---|---|---|---|
| Age (mean ± SD) | 59.3±16.3 | 51.6±13.2 | ns |
| Male (%) | 40.7% | 37.5% | ns |
| Caucasian (%) | 100% | 81.3% | 0.0454 |
| APACHE II Score (mean ± SD) | 11.1±5.9 | | |
| SOFA Score (mean ± SD) | 4.6±4.3 | | |
| % in Septic Shock | 55.6% | | |
| Time Post-Admission (hrs) | 6.1±5.6 | | |

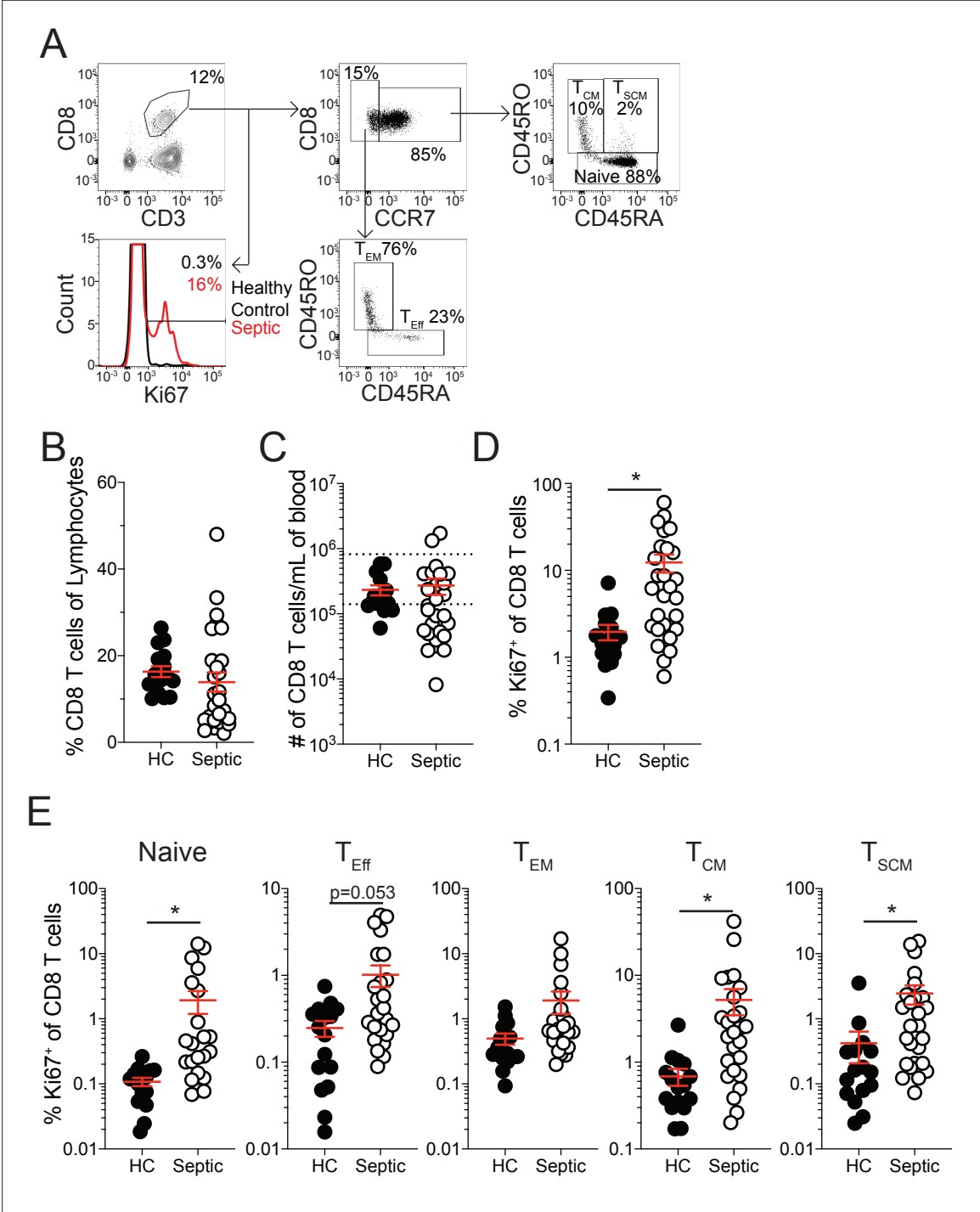

**Figure 1.** Increased proliferation among CD8 T cells of septic patients. (**A**) Representative gating for CD8 T cell subsets and Ki67 expression from healthy controls and septic patients (within 24 hr of hospital admission). (**B**) Frequency and (**C**) number of CD8 T cells among lymphocytes in healthy controls and septic patients. Dashed lines indicate the normal range for the number of CD8 T cells per mL of blood. (**D**) Frequency of Ki67 expressing CD8 T cells in healthy controls and septic patients. (**E**) Frequency Ki67 expressing cells among Naïve, Effector (T_Eff), Effector Memory (T_EM), Central Memory (T_CM), and Stem Cell Memory (T_SCM) CD8 T cells from healthy controls and septic patients. Data are representative of 2 independent experiments with 16–27 patients per group. *=p < 0.05. Error bars in represent standard error of the mean.

The online version of this article includes the following figure supplement(s) for figure 1:

**Source data 1.** Source data for *Figure 1*.

**Figure supplement 1.** Composition of total and proliferating CD8 T cells in healthy controls and septic hosts.

**Figure supplement 1—source data 1.** Source data for *Figure 1—figure supplement 1*.

This proliferation could either represent newly evoked effector CD8 T cell responses to the sepsis-inducing pathogens or homeostatic proliferation of surviving T cells induced by lymphopenic environment (*Cheung et al., 2009*; *Davenport et al., 2019*; *Jensen et al., 2018a*; *Unsinger et al., 2009*). To address this, the frequency of Ki67 expressing cells was evaluated between naïve (CCR7+CD45RA+CD45RO-), effector (T_{Eff}; CCR7-CD45RA+CD45RO-), effector memory (T_{EM}; CCR7-CD45RA-CD45RO+), central memory (T_{CM}; CCR7+CD45RA-CD45RO+), and stem cell memory (T_{SCM}; CCR7+CD45RA+CD45RO+) CD8 T cells (*Cieri et al., 2013*; *Sarkar et al., 2019*). If the proliferation was in response to the septic insult only T_{Eff} CD8 T cells should be prominently proliferating relative to healthy controls; however, there was only modest induction of Ki67 among T_{Eff} CD8 T cells (*Figure 1e*). Intriguingly, robust proliferation among naïve, T_{CM}, and T_{SCM} CD8 T cells was observed, suggesting proliferation may reflect numerical recovery after sepsis-induced lymphopenia (*Figure 1e*). Notably, there was not a significant increase in Ki67+ T_{EM} CD8 T cells from septic patients (*Figure 1e*). These data suggest that there is differential proliferation by memory CD8 T cell subsets in septic hosts. Given this differential proliferation by memory CD8 T cell subsets, an altered composition of the memory CD8 T cell pool would be anticipated after sepsis. Indeed, there was a modest, although not statistically different, increase in the frequency of both T_{CM} and T_{SCM} CD8 T cells in septic patients, relative to healthy controls, even at this early time point (*Figure 1—figure supplement 1a*). Importantly, when evaluating the representation of CD8 T cell subsets among Ki67-expressing CD8 T cells, T_{CM} and T_{SCM} CD8 T cells were not proportionally increased (*Figure 1—figure supplement 1b*), with T_{CM} being the most prominent among cells that recently proliferated. Collectively, these data suggest that sepsis may alter the composition of the memory CD8 T cell compartment due to intrinsic differences in the capacity of different memory CD8 T cell subsets to proliferate.

## Pre-existing memory 8 T cells numerically recover after sepsis

To further address how sepsis may alter the composition of the CD8 T cell compartment due to differential capacity memory CD8 T cell subsets to sense signals of the 'empty' environment and undergo homeostatic proliferation, we utilized a murine LCMV-infection model to establish memory CD8 T cells followed by a cecal ligation and puncture (CLP; *Figure 2a*). To facilitate resolution/analyses of the memory CD8 T cell compartment a physiologically relevant number of naïve Thy1.1+ TCR-Tg P14 CD8 T cells, specific for the GP_{33} epitope of LCMV, were adoptively transferred into Thy1.2+ recipient mice. Mice were then infected with LCMV-Arm, an acute infection which elicits a robust and well characterized memory CD8 T cell response (*Badovinac et al., 2007*). This system of memory generation and sepsis induction enables rigorous interrogation of a defined population of "pre-existing" memory CD8 T cells (memory cells that exist prior to sepsis induction) wherein both the time of the priming infection and septic event are known. Additionally, naïve and antigen-experienced (Ag-exp) CD8 T cells can be differentiated based on the expression of surrogate markers of activation CD8a and CD11a (naïve: CD8a^{hi}CD11a^{lo}; Ag-exp: CD8a^{lo}CD11a^{hi}) (*Rai et al., 2009*). This enables evaluation of endogenous naïve and Ag-exp CD8 T cells in addition to the Ag-exp P14 CD8 T cells (*Figure 2b*). Further depth of interrogation is achieved with memory P14 CD8 T cells, relative to the bulk antigen-experienced CD8 T cell population, given that memory P14 CD8 T cells are not specific for antigens evoked/released during the septic event, and the influence of sepsis on this discrete pre-existing memory CD8 T cell population delineates from potential 'secondary' antigen encounter and from potential and anticipated novel Ag-specific CD8 T cell responses to the septic event.

Following septic insult, the lymphopenic state impacted naïve and Ag-exp cells to the same degree (*Figure 2c*), as has been previously reported (*Condotta et al., 2013*; *Duong et al., 2014*; *Jensen et al., 2018a*). Importantly, the memory P14 CD8 T cells were similarly susceptible to sepsis-induced lymphopenia as the endogenous Ag-exp cells (*Figure 2c*). Additionally, there was induction of Ki67 expression by memory P14 CD8 T cells after sepsis (*Figure 2d and e*), demonstrating that the P14 CD8 T cells can be used to model the influence of sepsis on pre-existing memory CD8 T cells. When the number of memory P14 CD8 T cells per mL of blood was quantified, we observed numeric loss and recovery of P14 CD8 T cells in CLP hosts (*Figure 2f*), similar to prior reports of homeostatic proliferation following sepsis-induced lymphopenia (*Unsinger et al., 2009*). Thus, pre-existing memory CD8 T cells numerically recover with time after sepsis, potentially due to increased proliferation in response to the sepsis-induced lymphopenic environment.

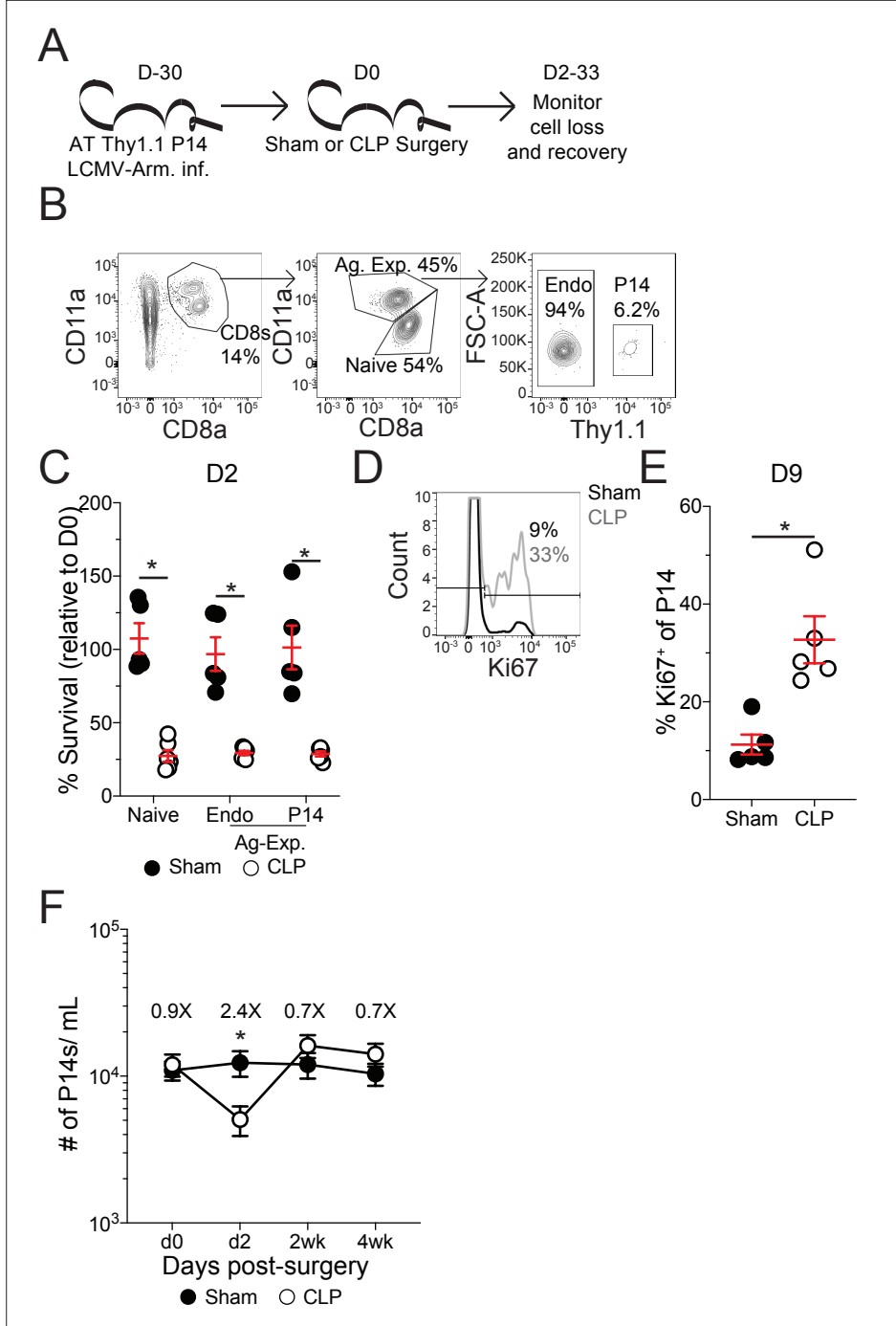

**Figure 2.** Pre-existing memory CD8 T cells numerically recover with time after sepsis. (**A**) Experimental Design: Antigen-experienced P14 chimeric mice were generated by adoptive transfer of $5 \times 10^3$ naïve Thy1.1+ TCR-transgenic P14 CD8 T cells to Thy1.2+ C57Bl/6 mice that were subsequently infected with LCMV-Armstrong (LCMV-Arm). Mice underwent Sham or CLP surgery 30 days after infection. The number of endogenous naïve, endogenous antigen-experienced, and antigen-experienced P14 CD8 T cells was monitored in the blood. (**B**) Representative gating for endogenous naïve, endogenous antigen-experienced, and antigen-experienced P14 CD8 T cells. (**C**) Percent survival of endogenous naïve, endogenous antigen-experienced, and antigen-experienced P14 CD8 T cells in the blood 2 days after either Sham or CLP surgery, relative to a pre-surgery bleed. (**D**) Representative gating of Ki67 on P14 CD8 T cells. (**E**) Frequency of Ki67-expressing P14 CD8 T cells in the blood of Sham and CLP hosts 9 days post-surgery. (**F**) The number of P14 CD8 T cells per mL of blood in Sham and CLP hosts prior to (d0), or 2 days (d2), 2 weeks (2 wk), and 4 weeks (4 wk) after surgery. Values above the bars indicate

*Figure 2 continued on next page*

*Figure 2 continued*

the fold difference (Sham/CLP) in the number of P14 CD8 T cells. (**C–E**) Are representative of 3 independent experiments with 5–6 mice per group. (**F**) Is cumulative from two independent experiments with 10–12 mice per group. *=p < 0.05. Error bars represent standard error of the mean.

The online version of this article includes the following figure supplement(s) for figure 2:

**Source data 1.** Source data for *Figure 2C*.

**Source data 2.** Source data for *Figure 2E*.

**Source data 3.** Source data for *Figure 2F*.

## Numeric recovery following sepsis increases the proportion of central memory CD8 T cells

To address how this numeric recovery may alter the composition of the memory T cell compartment phenotypic characterization of splenic memory P14 CD8 T cells from Sham and CLP mice was performed >30 days post-surgery (*Figure 3a*). Additionally, FlowSOM was utilized to cluster memory P14 CD8 T cells based on surface marker expression of CD8a, CD11a, Thy1.1, CD62L, KLRG1, CD127, CX3CR1, CXCR3, CD25, CD27, CD69, CD103, and CD122 in an unbiased manner (*Van Gassen et al., 2015*). Memory P14 CD8 T cells were similarly evaluated by tSNE analysis and FlowSOM-defined clusters were then projected into the tSNE (*Figure 3b and c*). Notably, Sham and CLP hosts had differential representation of two of the most prominent clusters (6 and 8) with cluster six being enriched in Sham P14 CD8 T cells and cluster eight in CLP P14 CD8 T cells (*Figure 3d and e*). Clusters 6 and 8 were then compared to define distinctions between Sham and CLP cells (*Figure 3g*). Memory P14 CD8 T cells enriched in Sham mice (cluster 6) were $CD62L^-KLRG1^+CD127^-CX3CR1^+CXCR3^{lo}$, while memory P14 CD8 T cells enriched in CLP mice (cluster 8) were $CD62L^+KLRG1^-CD127^+CX3CR1^-CXCR3^{lo/med}$ (*Figure 3h*). Clusters 6 and 8 therefore appear to define $T_{EM}$ and $T_{CM}$ CD8 T cells, respectively. Definition of these subsets was predominantly by the expression of CD62L, although the expression of KLRG1, CD127, CX3CR1, and CXCR3 conformed with the respective phenotypes as well (*Martin and Badovinac, 2018*). Thus, CLP P14 CD8 T cells are enriched for $T_{CM}$ with a reduced representation of $T_{EM}$, corresponding to the increased proliferation of CD8 $T_{CM}$ observed in septic patients (*Figure 1e*; *Figure 1—figure supplement 1b*).

$T_{CM}$ have a higher capacity to undergo homeostatic proliferation, relative to $T_{EM}$, which accounts for the gradual shift toward $T_{CM}$ with time after antigen encounter (*Martin and Badovinac, 2018*; *Martin et al., 2015*; *Wherry et al., 2003*). Therefore, to address whether the higher proliferative potential of $T_{CM}$ accounted for the shift to CD8 $T_{CM}$ following sepsis, Ki67 expression in splenic $T_{EM}$ and $T_{CM}$ P14 CD8 T cells was interrogated at various times after Sham or CLP surgery (*Figure 4a*). Indeed, both $T_{CM}$ and $T_{EM}$ proliferated in CLP hosts greater than their Sham counterparts, following lymphopenia (*Figure 4b*). However, P14 CD8 $T_{CM}$ cells proliferated more robustly than their $T_{EM}$ counterparts in CLP hosts. Importantly, P14 CD8 $T_{CM}$ cells proliferated more than their $T_{EM}$ counterparts in Sham hosts across all timepoints, consistent with prior reports of higher homeostatic proliferation among $T_{CM}$ cells (*Wherry et al., 2003*). To confirm the higher degree of proliferation in P14 CD8 $T_{CM}$ cells following sepsis, BrdU incorporation was evaluated over the course of a week beginning at D9 post-surgery, the timepoint at which differential proliferation had been observed by Ki67 expression (*Figure 4c*). Similar to the results with Ki67, elevated proliferation was observed in both $T_{CM}$ and $T_{EM}$ P14 CD8 T cells from CLP hosts, relative to Sham hosts (*Figure 4d*). Additionally, P14 CD8 $T_{CM}$ cells had higher incorporation of BrdU (relative to $T_{EM}$ counterparts) in both Sham and CLP hosts with P14 CD8 $T_{CM}$ cells from CLP hosts having the highest degree of BrdU incorporation. Similar results were also demonstrated in endogenous Ag-exp CD8 T cells reaffirming the findings in the TCR-Tg memory P14 CD8 T cells. This proliferative difference was further associated with an increase in the frequency of $T_{CM}$ among P14 CD8 T cells at D16 post-surgery, the time at which BrdU assessment was performed (*Figure 4e*). Additionally, a trending increase in the representation of $T_{CM}$ was observed among Ag-exp CD8 T cells in CLP hosts, relative to Sham hosts, at the same time in spite of potential novel effector CD8 T cell responses to the septic insult.

In addition to differential capacity to undergo homeostatic proliferation $T_{CM}$ and $T_{EM}$ have different localization throughout the body. $T_{CM}$ preferentially localizes to lymphatic tissue while $T_{EM}$ preferentially circulates and traverse non-lymphatic tissue (*Gerlach et al., 2016*; *Masopust et al., 2001*;

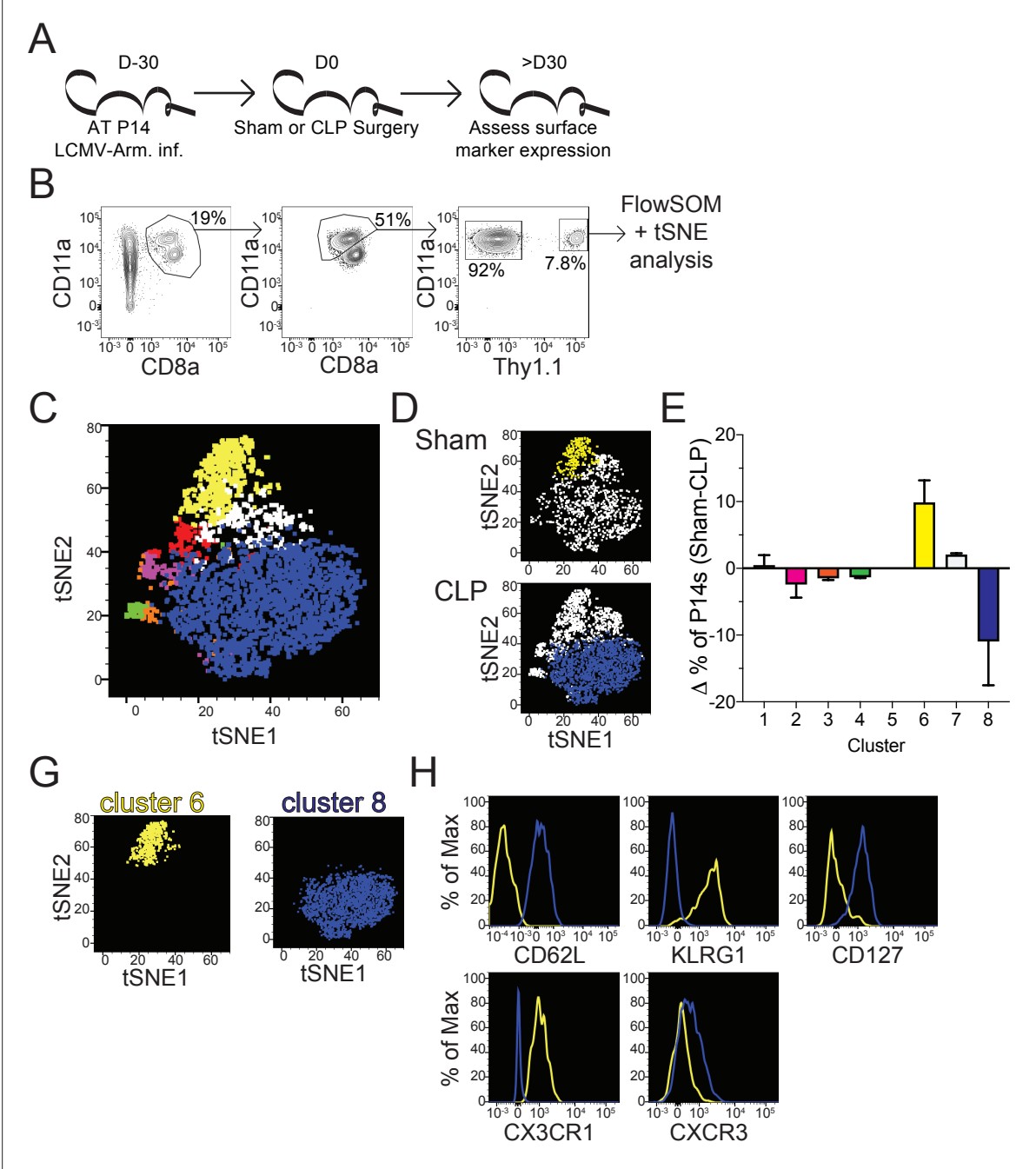

**Figure 3.** Sepsis alters the phenotypic composition of pre-existing memory CD8 T cells. (**A**) Experimental Design: Antigen-experienced P14 chimeric mice were generated by adoptive transfer of $5 \times 10^3$ naive Thy1.1[+] TCR-transgenic P14 CD8 T cells to Thy1.2[+] CD57Bl/6 mice that were subsequently infected with LCMV-Armstrong (LCMV-Arm). Mice underwent Sham or CLP surgery 30 days after infection. Phenotypic marker expression on P14 CD8 T cells was then assessed 30 days after surgery. (**B**) Representative antigen-experienced P14 CD8 T cells used in FlowSOM and tSNE analyses. (**C**) tSNE displaying FlowSOM defined clusters among P14 CD8 T cells based on surface marker expression of CD8a, CD11a, Thy1.1, CD62L, KLRG1, CD127, CX3CR1, CXCR3, CD25, CD27, CD69, CD103, and CD122. (**D**) Sham and CLP tSNE plots displaying clusters most robustly enriched in corresponding group. (**E**) Change (Δ) in the frequency of P14 CD8 T cells in each cluster (Sham-CLP); clusters biased toward Sham are >0, clusters biased toward CLP are <0. (**G**) tSNE plots displaying the clusters 6 (enriched in Sham hosts) and 8 (enriched in CLP hosts). (**H**) Surface expression of CD62L, KLRG1, CD127, CX3CR1, and CXCR3 comparing clusters 6 and 8. Data are representative of two independent experiments with 2–3 mice per group. Error bars indicate standard error of the mean.

The online version of this article includes the following figure supplement(s) for figure 3:

**Source data 1.** Source data for *Figure 3*.

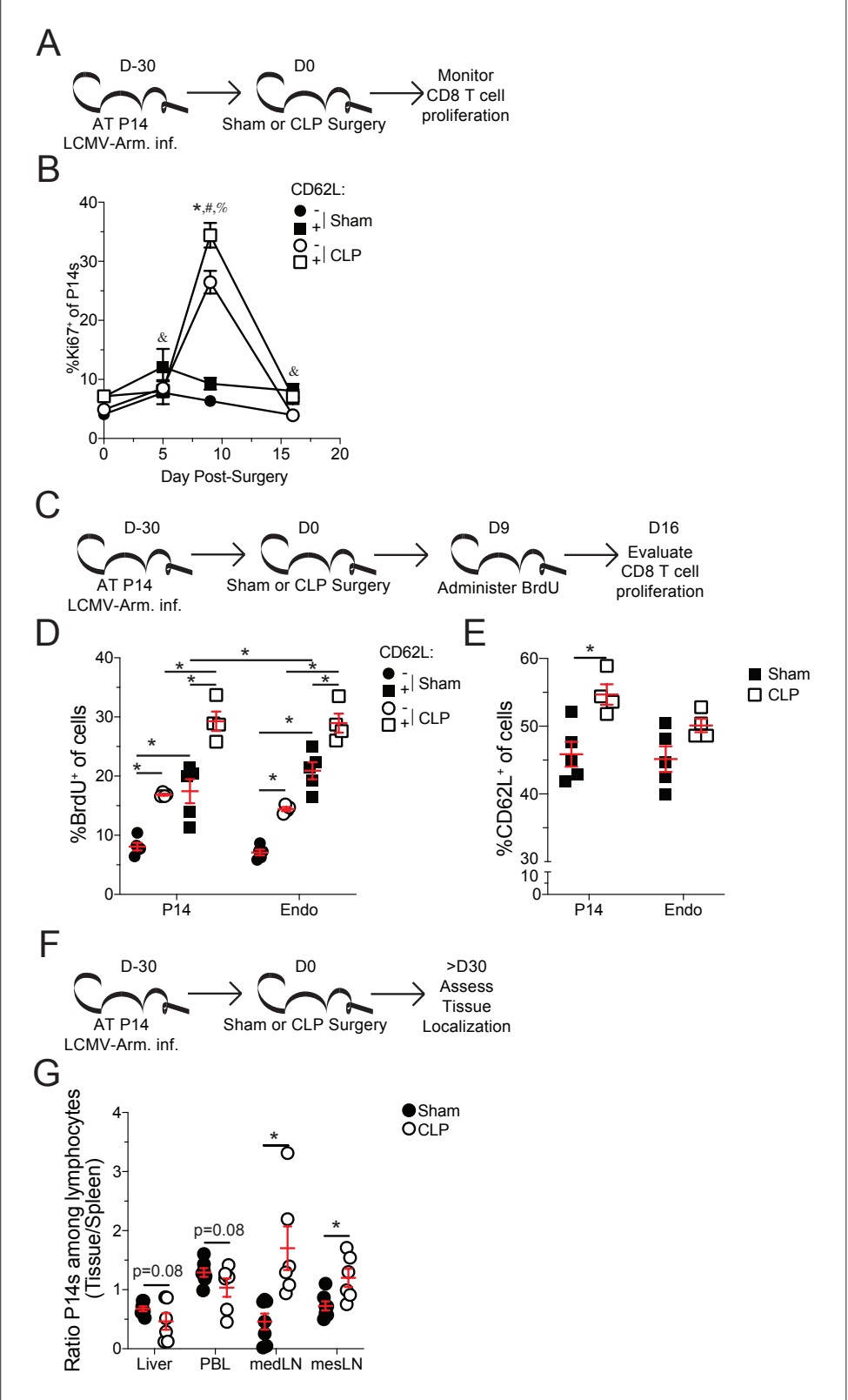

**Figure 4.** Central memory CD8 T cells more robustly proliferate after sepsis. (**A**) Experimental Design: Antigen-experienced P14 chimeric mice were generated by adoptive transfer of 5 × 10³ naive Thy1.1⁺ TCR-transgenic P14 CD8 T cells to Thy1.2⁺ C57Bl/6 mice that were subsequently infected with LCMV-Armstrong (LCMV-Arm). Mice underwent Sham or CLP surgery 30 days after infection. The frequency of Ki67 expressing central and effector

*Figure 4 continued on next page*

*Figure 4 continued*

memory P14 CD8 T cells was monitored in the spleen after surgery. (**B**) Frequency of Ki67 expressing cells among central (CD62L$^+$) and effector (CD62L$^-$) memory P14 CD8 T cells in Sham and CLP hosts prior to (d0) or 5-, 9-, and 16 days after surgery. *=p < 0.05 CD62L$^+$ v CD62L$^-$ CLP P14 CD8 T cells; $^\&$=p < 0.05 CD62L$^+$ v CD62L$^-$ Sham P14 CD8 T cells; $^\#$=p < 0.05 Sham v CLP CD62L$^+$ P14 CD8 T cells; $^\%$=p < 0.05 Sham v CLP CD62L$^-$ P14 CD8 T cells (**C**) Experimental Design: Antigen-experienced P14 chimeric mice were generated by adoptive transfer of 5 × 10$^3$ naïve Thy1.1$^+$ TCR-transgenic P14 CD8 T cells to Thy1.2$^+$ C57Bl/6 mice that were subsequently infected with LCMV-Arm. Mice underwent Sham or CLP surgery 30 days after infection followed by BrdU administration 9 days later. BrdU incorporation by central and effector memory endogenous and P14 CD8 T cells was assessed 7 days later. (**D**) Frequency of CD62L$^+$ and CD62L$^-$ memory P14 CD8 T cells and endogenous CD8 T cells that have incorporated BrdU. (**E**) Frequency of CD62L$^+$ P14 CD8 T cells and endogenous CD8 T cells 16 days after surgery. (**F**) Experimental Design: Antigen-experienced P14 chimeric mice were generated by adoptive transfer of 5 × 10$^3$ naive Thy1.1$^+$ TCR-transgenic P14CD8 T cells to Thy1.2$^+$ C57Bl/6 mice that were subsequently infected with LCMV-Arm. Mice underwent Sham or CLP surgery 30 days after infection. The frequency of P14 CD8 T cells among lymphocytes in the spleen, liver, PBL, mediastinal lymph node (medLN), and mesenteric lymph node (mesLN) was then determined 30 days after surgery. Preferential localization was determined by the ratio of P14 CD8 T cells in the tissues compared relative to the spleen. (**G**) Ratio of the frequency of P14 CD8 T cells among lymphocytes in the liver, PBL, medLN, and mesLN relative to the spleen. All data are representative of at least two independent experiments with 4–8 mice per group. *=p < 0.05. Error bars represent standard error of the mean.

The online version of this article includes the following figure supplement(s) for figure 4:

**Source data 1.** Source data for *Figure 4B*.

**Source data 2.** Source data for *Figure 4D and E*.

**Source data 3.** Source data for *Figure 4G*.

*Mueller et al., 2013*). To further address this shift in the representation of T$_{CM}$ and T$_{EM}$, the localization of memory P14 CD8 T cells was evaluated in the liver, peripheral blood lymphocytes (PBL), mediastinal lymph nodes (mLN), and mesenteric lymph nodes (mesLN) relative to the spleen (*Figure 4f*). Spleen was chosen as the baseline for comparison as it is a mixture of circulation with lymphatic tissue. PBL was chosen to emphasize circulating cells, while liver was chosen as a non-lymphatic tissue because it is a highly vascular tissue with direct contact to blood coming from the abdominal cavity and thus relevant to the septic insult. MedLN are the site of initial infection with LCMV-Arm following i.p. infection and is therefore relevant to the generation of the initial memory response (*Olson et al., 2012*), while mesLN drain the gut tissue and are relevant to CLP induction. Thus, if sepsis leads to a global shift toward central memory we expected to see a reduced proportion of memory P14 CD8 T cells in the liver and PBL and a greater proportion in the medLN and mesLN in CLP hosts, relative to Sham. Indeed, the ratio of P14 CD8 T cells among lymphocytes in the liver and PBL, relative to the spleen, had a trending reduction in CLP hosts, compared to Sham hosts (*Figure 4g*). Conversely, the ratio of P14 CD8 T cells among lymphocytes in the medLN and mesLN relative to the spleen, were significantly increased in CLP hosts, compared to Sham hosts. These data demonstrate differential localization of CD8 T cells in Sham and CLP hosts corresponding to the change in the representation of T$_{EM}$ and T$_{CM}$. Cumulatively, the data in *Figure 4* demonstrate that preferential proliferation by T$_{CM}$ alters the composition and localization of pre-existing memory CD8 T cells after sepsis. Thus, pre-existing differences in the biology of central and effector memory T cells are the underlying mechanism by which central memory CD8 T cells become over-represented in pre-existing memory populations after sepsis.

## Sepsis leads to long-term changes in memory CD8 T cell transcription and chromatin accessibility

Beyond localization T$_{CM}$ and T$_{EM}$ have differential functions mediated by discrete transcriptional and epigenetic landscapes (*Chang et al., 2014*; *Kaech and Cui, 2012*; *Milner et al., 2020*). Therefore, to address how the sepsis-induced changes in the composition of pre-existing memory CD8 T cells may alter the overall transcriptional regulation of memory CD8 T cells RNA-sequencing was performed on memory P14 CD8 T cells from Sham and CLP hosts both 1- and 31 days post-surgery (*Figure 5a*). Numerous transcriptional differences between the four groups were identified (*Figure 5b–d*). Notably, when evaluated by principal component analysis (PCA) there was clear distinction between the Sham and CLP groups at each timepoint (*Figure 5b*); however, this distinction narrowed at D31 relative to

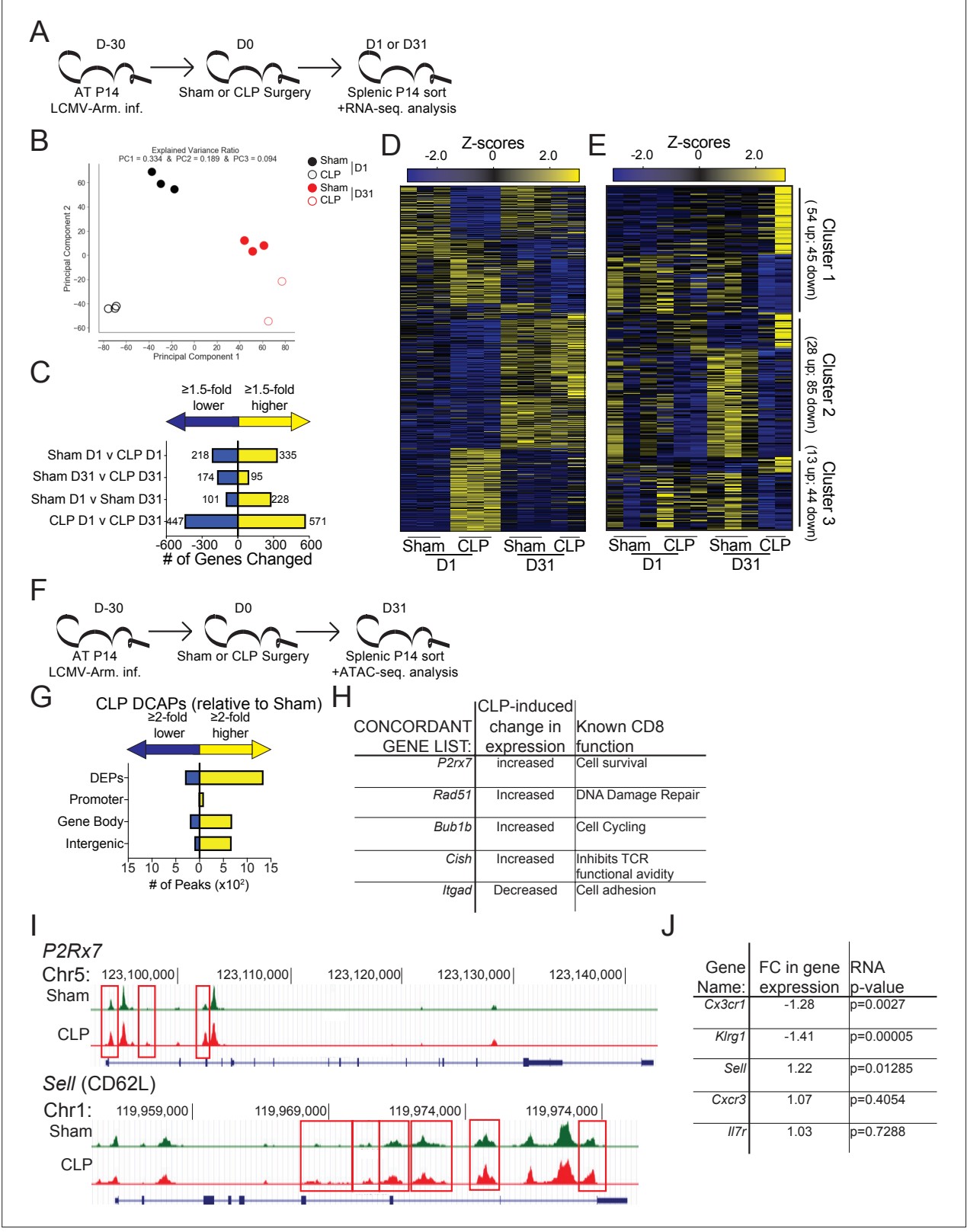

**Figure 5.** Sepsis alters the gene expression and chromatin accessibility of pre-existing memory CD8 T cells. (**A**) Experimental Design: Antigen-experienced P14 chimeric mice were generated by adoptive transfer of 5 × 10³ naive Thy1.1⁺ TCR-transgenic P14 CD8 T cells to Thy1.2⁺ C57Bl/6 mice that were subsequently infected with LCMV-Arm. Mice underwent Sham or CLP surgery 30 days after infection. Splenic P14 CD8 T cells were FACS-sorted one or 31 after surgery for RNA extraction. P14 CD8 T cells were isolated from 3 D1-Sham hosts, 3 D1-CLP hosts, 3 D31-Sham hosts, and 2 D31-

*Figure 5 continued on next page*

*Figure 5 continued*

CLP hosts. (**B**) Principal Component analysis of P14 CD8 T cells from Sham and CLP hosts either 1- or 31 days post-surgery. (**C**) Number of statistically significant gene changes as a result of indicated comparisons. (**D**) Gene expression heatmap of genes with statistically significant changes (fold change >1.5, p < 0.05) as a result of any comparison. (**E**) Gene expression heatmap of genes with statistically significant changes (fold change >1.5, p < 0.05) between D31 Sham and CLP P14 CD8 T cells. Clusters were consecutively defined by similar expressional changes in: D1 to D31 Sham P14 CD8 T cells and D31 Sham to CLP P14 CD8 T cells [Cluster 1], D1 Sham to CLP P14 CD8 T cells and D31 Sham to CLP P14 CD8 T cells [Cluster 2], and non-defined by prior categorization [Cluster 3] (**F**) Experimental Design: Antigen-experienced P14 chimeric mice were generated by adoptive transfer of 5 × $10^3$ naive Thy1.1$^+$ TCR-transgenic P14 CD8 T cells to Thy1.2$^+$ C57Bl/6 mice that were subsequently infected with LCMV-Arm. Mice underwent Sham or CLP surgery 30 days after infection. Splenic P14 CD8 T cells were FACS-sorted 31 days after surgery for assessment of chromatin accessibility. P14 CD8 T cells were isolated from 2 D31-Sham hosts and 3 D31-CLP hosts. (**G**) Total number of differential chromatin accessibility peaks (DCAPs, fold change >2 p < 0.05) and delineation of those within either a promoter, gene body, or intergenic regions assigned to the most proximal to a transcription start site. (**H**) List of genes whose change in transcript is concordant with changes in chromatin accessibility along with the relative change and known function in CD8 T cells. (**I**) Example of differentially expressed peaks (indicated by the red box) within the *P2R×7* and *Sell* gene loci from representative Sham and CLP P14s. (**J**) List of genes whose expression defined the phenotypically distinct populations between Sham and CLP P14 CD8 T cells in *Figure 3* alongside their fold change in transcript and the p-value associated with that fold-change.

The online version of this article includes the following figure supplement(s) for figure 5:

**Source data 1.** Source data for *Figure 5C and D*.

**Source data 2.** Source data for *Figure 5E*.

D1 potentially reflecting some degree of recovery from the initial the cytokine storm. Additionally, D1 and D31 were distinct within their respective surgical groups recapitulating prior literature demonstrating how memory changes with time after antigen encounter (*Davenport et al., 2019*; *Martin et al., 2017*). Importantly, numerous gene changes (269 total; 174 down, 95up) were present at D31 post-surgery in CLP hosts, relative to Sham controls (*Figure 5c and d*). To understand how these lasting transcriptional changes related to the septic insult, the significantly different genes expressed at D31 by Sham and CLP memory P14 CD8 T cells were clustered into three groups (*Figure 5e*). Given our observation of faster transition to central memory by CD8 T cells after sepsis the first cluster of genes identified were those that were similarly changed when comparing D1 to D31 Sham and D31 Sham to CLP P14 CD8 T cells. These changes constituted 99 of the 269 transcriptional (*Table 2*) differences between Sham and CLP memory P14 CD8 T cells at D31 and validate the observations in the prior figures of more rapid adoption of time-dependent changes in memory (i.e. conversion to central memory) (e.g. changes in clusters 6 and 8 of *Figure 3*). The remaining genes were then evaluated for the presence of a sepsis-induced transcriptional 'scar' to delineate conserved changes as a result of the septic event. These changes constituted the second cluster and were identified by the similar transcriptional changes for D1 Sham to CLP and D31 Sham to CLP. This sepsis-induced 'scar' constituted 113 of the 269 gene changes observed (*Table 2*) and demonstrates that some of the transcriptional changes evoked early after sepsis persist. Finally, there remained a third cluster of 57 gene changes (*Table 2*) that were neither associated with time-dependent changes in CD8 T cell memory nor were they associated with the early sepsis induced changes to memory CD8 T cells. Thus, novel transcriptional changes also arise in memory CD8 T cells during the post-septic environment. Summarily, these data demonstrate that sepsis leads to lasting changes in the transcriptional landscape of memory CD8 T cells. These changes are associated with the more rapid acquisition of time-dependent changes by memory CD8 T cells in CLP hosts, a sepsis-induced transcriptional scar, and novel transcriptional changes acquired in the post-septic environment.

To address how these transcriptional changes may be the result of an altered epigenetic landscape, chromatin accessibility was assessed in Sham and CLP P14 CD8 T cells by ATAC-sequencing at D31 post-surgery (*Figure 5f*). While 1646 peaks were differentially expressed, the changes observed were predominantly more peaks (more accessibility) in CLP hosts (*Figure 5g*). Of these the majority were either within a gene body or intergenic regions assigned to the nearest a transcription start site. Significant changes in gene expression were then compared with DCAPs to establish whether there was concordance between the gene changes observed and the accessibility of the chromatin. Indeed, there were genes whose chromatin accessibility and transcription is concordant. Importantly, these concordant genes identified potentially relevant changes in CD8 T cell function (*Figure 5h and i*). Among these *P2r×7*, *Rad51*, and *Bub1b* all have prior association with CD8 T cell survival, DNA damage repair, and cell cycling (*Baek et al., 2003*; *Borges da Silva et al., 2018*; *Yamamoto et al.,*

**Table 2.** Gene clusters.

| gene_id | Relative FC D32 sham vs CLP | p_value_ | Cluster # |
|---|---|---|---|
| Cdc6 | 4.03064298 | 0.00005 | Cluster 1 |
| Tppp3 | 4.88825654 | 0.00005 | Cluster 1 |
| Neil3 | 3.66496647 | 0.00005 | Cluster 1 |
| Hist1h1e | 3.76231332 | 0.00005 | Cluster 1 |
| Mcm10 | 3.47460746 | 0.00005 | Cluster 1 |
| Ttc8 | 3.38918878 | 0.00005 | Cluster 1 |
| Stmn1 | 2.06406728 | 0.0508 | Cluster 1 |
| Gpr34 | 3.41697573 | 0.00005 | Cluster 1 |
| Ppp2r2c | 1.83454208 | 0.0437 | Cluster 1 |
| Kntc1 | 2.99415727 | 0.00005 | Cluster 1 |
| Bfsp1 | 1.60516322 | 0.03045 | Cluster 1 |
| Birc5 | 2.8137029 | 0.00005 | Cluster 1 |
| Ccdc136 | 2.28917856 | 0.0344 | Cluster 1 |
| Gm5124 | 1.60472157 | 0.03095 | Cluster 1 |
| Ccnb2 | 2.49461988 | 0.00005 | Cluster 1 |
| Apol7b | 2.6692603 | 0.00005 | Cluster 1 |
| Tktl1 | 1.9334132 | 0.02895 | Cluster 1 |
| Dtl | 2.59828557 | 0.00005 | Cluster 1 |
| Pask | 2.7640812 | 0.00005 | Cluster 1 |
| Crip2 | 2.52029044 | 0.0004 | Cluster 1 |
| Clspn | 2.45116907 | 0.00015 | Cluster 1 |
| Mki67 | 2.61164716 | 0.00005 | Cluster 1 |
| Fam64a | 2.65731637 | 0.0006 | Cluster 1 |
| 2810408I11Rik | 1.5215239 | 0.06725 | Cluster 1 |
| Rad51ap1 | 1.63064417 | 0.00235 | Cluster 1 |
| Tnfsf4 | 2.37518519 | 0.0009 | Cluster 1 |
| E2f1 | 2.34354094 | 0.031 | Cluster 1 |
| Cep55 | 2.4930123 | 0.0004 | Cluster 1 |
| Morn3 | 2.12074329 | 0.00035 | Cluster 1 |
| Aurkb | 2.4030389 | 0.00005 | Cluster 1 |
| Hist2h2bb | 2.02820011 | 0.0464 | Cluster 1 |
| Exo1 | 2.36813282 | 0.00005 | Cluster 1 |
| Fcrlb | 1.52457593 | 0.04225 | Cluster 1 |
| Tmem176a | 1.59714799 | 0.0361 | Cluster 1 |
| Socs2 | 2.02128114 | 0.00155 | Cluster 1 |
| Ncapg2 | 1.97320931 | 0.00745 | Cluster 1 |
| Klra9 | 2.11296665 | 0.00005 | Cluster 1 |
| Chek1 | 1.59507471 | 0.05475 | Cluster 1 |

*Table 2 continued on next page*

*Table 2 continued*

| gene_id | Relative FC D32 sham vs CLP | p_value_ | Cluster # |
|---|---|---|---|
| Rad51 | 1.58587164 | 0.00005 | Cluster 1 |
| Dscc1 | 1.74867599 | 0.00015 | Cluster 1 |
| Bzrap1 | 1.61419626 | 0.00005 | Cluster 1 |
| Cd300e | 1.519849 | 0.07755 | Cluster 1 |
| Gm1720 | 1.94574412 | 0.0019 | Cluster 1 |
| Brca1 | 1.5604888 | 0.05435 | Cluster 1 |
| Gm14124 | 1.64708499 | 0.03685 | Cluster 1 |
| Shcbp1 | 1.79205262 | 0.00005 | Cluster 1 |
| Nebl | 1.8053468 | 0.04325 | Cluster 1 |
| Ckap2l | 1.73019782 | 0.0003 | Cluster 1 |
| Cdkn2a | 1.82961741 | 0.00085 | Cluster 1 |
| Phlda3 | 1.63185741 | 0.03825 | Cluster 1 |
| Adgre4 | 1.57646266 | 0.00005 | Cluster 1 |
| Klri2 | 1.56007674 | 0.0011 | Cluster 1 |
| Mmp25 | 1.54571785 | 0.04745 | Cluster 1 |
| Nenf | 1.5307705 | 0.0182 | Cluster 1 |
| Igf1 | −1.524316 | 0.01 | Cluster 1 |
| Rac3 | −1.5034855 | 0.07065 | Cluster 1 |
| Trf | −1.6496992 | 0.00015 | Cluster 1 |
| Chaf1a | −1.6002474 | 0.0011 | Cluster 1 |
| Orc1 | −1.6245667 | 0.04185 | Cluster 1 |
| Fignl1 | −1.7257028 | 0.0216 | Cluster 1 |
| D430020J02Rik | −1.6263897 | 0.0051 | Cluster 1 |
| Cenph | −1.8078839 | 0.0047 | Cluster 1 |
| Blvrb | −1.8434164 | 0.0457 | Cluster 1 |
| Cpne7 | −1.5149666 | 0.02655 | Cluster 1 |
| Psrc1 | −1.7535019 | 0.00145 | Cluster 1 |
| Uhrf1 | −1.5739852 | 0.0013 | Cluster 1 |
| Plbd1 | −1.7851892 | 0.0021 | Cluster 1 |
| Rgs12 | −1.8455643 | 0.01885 | Cluster 1 |
| Hpgd | −1.8522078 | 0.06025 | Cluster 1 |
| P2rx7 | −2.0648257 | 0.01595 | Cluster 1 |
| Bub1b | −2.2247142 | 0.0223 | Cluster 1 |
| 4833418N02Rik | −2.3048295 | 0.011 | Cluster 1 |
| Ube2c | −1.621509 | 0.0124 | Cluster 1 |
| Cadm1 | −2.3409109 | 0.00015 | Cluster 1 |
| Tyrobp | −2.4355709 | 0.00335 | Cluster 1 |
| Jup | −2.4842838 | 0.0023 | Cluster 1 |

*Table 2 continued on next page*

*Table 2 continued*

| gene_id | Relative FC D32 sham vs CLP | p_value_ | Cluster # |
|---|---|---|---|
| Pyroxd2 | –2.2264265 | 0.0282 | Cluster 1 |
| Gins1 | –1.8339216 | 0.00005 | Cluster 1 |
| Gm4013 | –2.7505116 | 0.01255 | Cluster 1 |
| Axl | –2.6625156 | 0.00005 | Cluster 1 |
| Nr6a1 | –2.729109 | 0.00125 | Cluster 1 |
| Hspa2 | –2.796573 | 0.00005 | Cluster 1 |
| Spry2 | –2.7405966 | 0.00005 | Cluster 1 |
| Mpeg1 | –2.8281527 | 0.00005 | Cluster 1 |
| Ticrr | –2.9418017 | 0.00005 | Cluster 1 |
| Plxdc1 | –2.9468017 | 0.00005 | Cluster 1 |
| Ly86 | –3.2898687 | 0.0019 | Cluster 1 |
| Cd302 | –3.3417875 | 0.00005 | Cluster 1 |
| C6 | –3.3367417 | 0.0003 | Cluster 1 |
| Slc37a2 | –3.3489759 | 0.00005 | Cluster 1 |
| Gm2011 | –3.4732831 | 0.00005 | Cluster 1 |
| H2-T3 | –2.0828371 | 0.00935 | Cluster 1 |
| Wnt2b | –3.5079301 | 0.0004 | Cluster 1 |
| Clec12a | –3.0562022 | 0.00895 | Cluster 1 |
| Mmp17 | –4.1251312 | 0.00005 | Cluster 1 |
| Ncaph | –4.1603359 | 0.0417 | Cluster 1 |
| Kcnj10 | –4.2878368 | 0.00005 | Cluster 1 |
| Cd163 | –4.2866184 | 0.0001 | Cluster 1 |
| Il11 | –5.666901 | 0.0001 | Cluster 1 |
| Syk | 4.80557812 | 0.0603 | Cluster 2 |
| 9030619P08Rik | 2.73623094 | 0.0353 | Cluster 2 |
| Prss16 | 2.57604894 | 0.00005 | Cluster 2 |
| Tcf4 | 2.52783097 | 0.0069 | Cluster 2 |
| Itgad | 2.32061951 | 0.00005 | Cluster 2 |
| Abcc3 | 2.47441931 | 0.0143 | Cluster 2 |
| Hfe | 1.91152093 | 0.00065 | Cluster 2 |
| Fcgr3 | 1.94948898 | 0.00005 | Cluster 2 |
| Rab3il1 | 2.00253853 | 0.00255 | Cluster 2 |
| Lrp1 | 2.06172226 | 0.06585 | Cluster 2 |
| Cd5l | 1.6117422 | 0.0645 | Cluster 2 |
| Mir2861 | 1.93355795 | 0.0162 | Cluster 2 |
| Il18 | 1.64165162 | 0.07875 | Cluster 2 |
| Mir155hg | 1.9632828 | 0.0003 | Cluster 2 |
| Irf4 | 1.83011207 | 0.00085 | Cluster 2 |

*Table 2 continued on next page*

*Table 2 continued*

| gene_id | Relative FC D32 sham vs CLP | p_value_ | Cluster # |
|---|---|---|---|
| Cmklr1 | 1.71253206 | 0.00655 | Cluster 2 |
| Mt3 | 1.61299839 | 0.00005 | Cluster 2 |
| Cd163l1 | 1.68200967 | 0.0028 | Cluster 2 |
| Palm | 1.69858804 | 0.0309 | Cluster 2 |
| Hmox1 | 1.8719138 | 0.0014 | Cluster 2 |
| Mertk | 1.69725108 | 0.05425 | Cluster 2 |
| Esm1 | 1.55538421 | 0.08875 | Cluster 2 |
| Lrrc25 | 1.53588266 | 0.01125 | Cluster 2 |
| Lgmn | 1.53366352 | 0.0421 | Cluster 2 |
| Mafb | 1.59909428 | 0.00005 | Cluster 2 |
| Havcr2 | 1.61859719 | 0.00005 | Cluster 2 |
| Epb4.1l3 | 1.64271249 | 0.0257 | Cluster 2 |
| Siglece | 1.58382507 | 0.0091 | Cluster 2 |
| Prr5 | –1.5418463 | 0.05615 | Cluster 2 |
| Pla2g7 | –1.5675528 | 0.00005 | Cluster 2 |
| Dusp14 | –1.6724627 | 0.02015 | Cluster 2 |
| Tgm2 | –1.6678575 | 0.04595 | Cluster 2 |
| Riiad1 | –1.5429946 | 0.00005 | Cluster 2 |
| Lilrb4a | –1.6883324 | 0.00685 | Cluster 2 |
| Ninj2 | –1.5669531 | 0.08315 | Cluster 2 |
| Cish | –1.6443416 | 0.0508 | Cluster 2 |
| Cenpe | –1.8650444 | 0.07995 | Cluster 2 |
| Cenpm | –1.5922104 | 0.00155 | Cluster 2 |
| Tpx2 | –1.5558403 | 0.05695 | Cluster 2 |
| Oip5 | –1.7861225 | 0.00005 | Cluster 2 |
| Cdca7 | –1.5875434 | 0.04035 | Cluster 2 |
| Ckap2 | –1.6603956 | 0.00425 | Cluster 2 |
| Ncapg | –1.8932803 | 0.019 | Cluster 2 |
| Ssc4d | –1.9973193 | 0.0063 | Cluster 2 |
| Stkld1 | –1.8746276 | 0.04135 | Cluster 2 |
| Cdca8 | –1.6650517 | 0.0087 | Cluster 2 |
| Cdc45 | –1.9715619 | 0.00405 | Cluster 2 |
| Lrp11 | –1.9542283 | 0.0003 | Cluster 2 |
| Mcm5 | –1.965633 | 0.012 | Cluster 2 |
| Cks1b | –2.0197826 | 0.0137 | Cluster 2 |
| Apitd1 | –1.614175 | 0.01045 | Cluster 2 |
| Spc24 | –2.1192151 | 0.00145 | Cluster 2 |
| Serpine2 | –2.0231734 | 0.00005 | Cluster 2 |

*Table 2 continued on next page*

*Table 2 continued*

| gene_id | Relative FC D32 sham vs CLP | p_value_ | Cluster # |
|---|---|---|---|
| Brip1 | –1.6461742 | 0.092 | Cluster 2 |
| Pole | –2.2240204 | 0.0253 | Cluster 2 |
| Lig1 | –1.7583399 | 0.0015 | Cluster 2 |
| Cenpn | –1.7057202 | 0.00005 | Cluster 2 |
| Gm19434 | –2.012432 | 0.0005 | Cluster 2 |
| Carns1 | –2.0274411 | 0.0205 | Cluster 2 |
| Mpp2 | –1.5418367 | 0.00775 | Cluster 2 |
| Mustn1 | –2.061708 | 0.0017 | Cluster 2 |
| Rn45s | –2.10963 | 0.07705 | Cluster 2 |
| Sema6b | –2.2547383 | 0.0119 | Cluster 2 |
| Cfp | –2.0456935 | 0.0001 | Cluster 2 |
| App | –2.211262 | 0.00935 | Cluster 2 |
| Car9 | –2.0062063 | 0.02255 | Cluster 2 |
| 1700102P08Rik | –2.4325509 | 0.0268 | Cluster 2 |
| Snhg10 | –2.345361 | 0.00015 | Cluster 2 |
| Lima1 | –2.3391753 | 0.07585 | Cluster 2 |
| Selm | –2.0294095 | 0.00265 | Cluster 2 |
| Slc41a3 | –2.1887998 | 0.0001 | Cluster 2 |
| Src | –2.6392902 | 0.00015 | Cluster 2 |
| Miat | –2.5487856 | 0.0109 | Cluster 2 |
| Cd79a | –2.5900501 | 0.00055 | Cluster 2 |
| Plxnb2 | –2.9874612 | 0.00005 | Cluster 2 |
| Pla2g2d | –1.529167 | 0.0268 | Cluster 2 |
| Lyz2 | –2.9362402 | 0.04775 | Cluster 2 |
| Cdk3-ps | –3.0692156 | 0.00265 | Cluster 2 |
| Mir6236 | –2.5029162 | 0.0004 | Cluster 2 |
| Smagp | –1.9662967 | 0.00005 | Cluster 2 |
| AF251705 | –2.8636993 | 0.0226 | Cluster 2 |
| Gfra2 | –2.8780666 | 0.00005 | Cluster 2 |
| Tmem91 | –3.2171875 | 0.00005 | Cluster 2 |
| Pld4 | –1.5079912 | 0.02785 | Cluster 2 |
| Itgb5 | –1.6713455 | 0.00225 | Cluster 2 |
| Treml4 | –3.2826024 | 0.0001 | Cluster 2 |
| Cd14 | –2.7112347 | 0.00005 | Cluster 2 |
| Marcks | –3.1963602 | 0.00005 | Cluster 2 |
| Cmbl | –2.6693158 | 0.00005 | Cluster 2 |
| Klra3 | –2.5417462 | 0.03455 | Cluster 2 |
| Ctsh | –3.5891694 | 0.00565 | Cluster 2 |

*Table 2 continued on next page*

*Table 2 continued*

| gene_id | Relative FC D32 sham vs CLP | p_value_ | Cluster # |
|---|---|---|---|
| Klra8 | –3.3087585 | 0.0001 | Cluster 2 |
| Cd81 | –3.3276415 | 0.00005 | Cluster 2 |
| C1qb | –3.7316146 | 0.00005 | Cluster 2 |
| Aif1 | –3.2695906 | 0.01525 | Cluster 2 |
| Bank1 | –3.0876517 | 0.002 | Cluster 2 |
| C1qc | –3.0727492 | 0.0001 | Cluster 2 |
| Apoe | –3.5282676 | 0.00005 | Cluster 2 |
| Clec4a3 | –2.9215824 | 0.00015 | Cluster 2 |
| Tgfbi | –3.5641054 | 0.00005 | Cluster 2 |
| Mrc1 | –3.6479097 | 0.03025 | Cluster 2 |
| Sirpa | –4.0829446 | 0.00005 | Cluster 2 |
| Clec1b | –1.7537681 | 0.0004 | Cluster 2 |
| Klra14-ps | –3.8984753 | 0.00005 | Cluster 2 |
| Ccr3 | –2.5323907 | 0.06515 | Cluster 2 |
| C1qa | –3.5880749 | 0.00005 | Cluster 2 |
| Vcam1 | –3.6726209 | 0.0011 | Cluster 2 |
| Tbxas1 | –4.4023688 | 0.00005 | Cluster 2 |
| Csf1r | –4.0805398 | 0.00005 | Cluster 2 |
| Fcna | –3.3668046 | 0.00005 | Cluster 2 |
| Adgre1 | –3.7926073 | 0.00005 | Cluster 2 |
| Adamdec1 | –4.9789792 | 0.00005 | Cluster 2 |
| Tnfrsf8 | –2.2700751 | 0.0008 | Cluster 2 |
| Aldh2 | 2.4260512 | 0.00025 | Cluster 3 |
| Slc40a1 | 2.40044187 | 0.0057 | Cluster 3 |
| Zfp385a | 2.16534965 | 0.06435 | Cluster 3 |
| Spag5 | 1.88610893 | 0.00005 | Cluster 3 |
| Nusap1 | 1.84390075 | 0.09965 | Cluster 3 |
| B9d1 | 1.78604818 | 0.0031 | Cluster 3 |
| Top2a | 1.74249558 | 0.0044 | Cluster 3 |
| Alox5ap | 1.64982614 | 0.09075 | Cluster 3 |
| Sgol2a | 1.61420186 | 0.05365 | Cluster 3 |
| Cdk1 | 1.59276453 | 0.0489 | Cluster 3 |
| Pla2g4b | 1.58954405 | 0.04465 | Cluster 3 |
| Fam174b | 1.54600394 | 0.00895 | Cluster 3 |
| Spc25 | 1.51665085 | 0.0962 | Cluster 3 |
| Ppp1r13l | –1.5091602 | 0.0399 | Cluster 3 |
| Neto2 | –1.5160875 | 0.0106 | Cluster 3 |
| Eif3j2 | –1.5188243 | 0.03625 | Cluster 3 |

*Table 2 continued on next page*

*Table 2 continued*

| gene_id | Relative FC D32 sham vs CLP | p_value_ | Cluster # |
|---|---|---|---|
| Gm28042 | −1.5257346 | 0.02945 | Cluster 3 |
| Gm4532 | −1.5651414 | 0.05395 | Cluster 3 |
| Sowahc | −1.5972355 | 0.09105 | Cluster 3 |
| Tnfrsf21 | −1.6215854 | 0.04005 | Cluster 3 |
| Gm3435 | −1.6321142 | 0.01435 | Cluster 3 |
| Zfp112 | −1.6371301 | 0.04985 | Cluster 3 |
| Nr1h3 | −1.6602966 | 0.07725 | Cluster 3 |
| 2810468N07Rik | −1.6797191 | 0.0173 | Cluster 3 |
| Hck | −1.6894024 | 0.04875 | Cluster 3 |
| Pth1r | −1.7203976 | 0.06865 | Cluster 3 |
| Tagln3 | −1.7598763 | 0.05825 | Cluster 3 |
| Hist1h4d | −1.7606181 | 0.0688 | Cluster 3 |
| Tubb3 | −1.803737 | 0.00005 | Cluster 3 |
| Klre1 | −1.8122236 | 0.02795 | Cluster 3 |
| Spi1 | −1.8248628 | 0.0498 | Cluster 3 |
| Fcgr4 | −1.8961012 | 0.01005 | Cluster 3 |
| Mrgpre | −1.9259777 | 0.00455 | Cluster 3 |
| Chrne | −1.9429018 | 0.03985 | Cluster 3 |
| Tctex1d2 | −1.9516374 | 0.00005 | Cluster 3 |
| Sdc3 | −1.9572554 | 0.0026 | Cluster 3 |
| Tlr7 | −1.9874423 | 0.0019 | Cluster 3 |
| Slc11a1 | −2.0856253 | 0.001 | Cluster 3 |
| Gzma | −2.102273 | 0.00005 | Cluster 3 |
| Cpsf4l | −2.1529279 | 0.0146 | Cluster 3 |
| Clec4a1 | −2.2106337 | 0.0132 | Cluster 3 |
| Fcer1g | −2.2631622 | 0.0002 | Cluster 3 |
| Ncf2 | −2.2765987 | 0.00205 | Cluster 3 |
| Slpi | −2.2838533 | 0.00005 | Cluster 3 |
| Cd244 | −2.4303599 | 0.0011 | Cluster 3 |
| Ptgs1 | −2.5181601 | 0.0003 | Cluster 3 |
| Cybb | −2.7556258 | 0.0001 | Cluster 3 |
| Matk | −3.0595087 | 0.00005 | Cluster 3 |
| Ifitm2 | −3.1387932 | 0.01575 | Cluster 3 |
| Cdc20b | −3.2338662 | 0.00705 | Cluster 3 |
| Msc | −3.4433671 | 0.01 | Cluster 3 |
| Clec4n | −3.4851723 | 0.00015 | Cluster 3 |
| Rgl1 | −3.5597601 | 0.0001 | Cluster 3 |
| Spic | −3.6767727 | 0.00005 | Cluster 3 |

*Table 2 continued on next page*

*Table 2 continued*

| gene_id | Relative FC D32 sham vs CLP | p_value_ | Cluster # |
|---|---|---|---|
| Hebp1 | −3.7616875 | 0.00025 | Cluster 3 |
| Hist1h3e | −4.1043962 | 0.04245 | Cluster 3 |
| Lrg1 | −5.383073 | 0.0033 | Cluster 3 |

*1996*). Thus, the increase in the expression of these genes in CLP P14 CD8 T cells likely reflects the numerical recovery after sepsis and general shift toward T$_{CM}$. Conversely, the increased expression of *Cish* and decreased expression of *Itgad*, which inhibit TCR functional avidity (*Palmer et al., 2015*) and promote cell adhesion (*Siegers et al., 2017*), respectively, demonstrate the function of surviving CD8 T cells may be compromised or altered. This is particularly interesting given that the TCR is itself fixed in these populations thus the variations are not attributable to changes in the composition of the TCR repertoire, a finding that would not be obvious in a polyclonal TCR population. To relate these findings back to the phenotypic differences observed previously, the chromatin accessibility within the *Sell* locus (which encodes CD62L) was compared between P14 CD8 T cells from Sham and CLP hosts (*Figure 5i*). Critically, there was increased accessibility in the CD62L locus of P14 CD8 T cells from CLP hosts, relative to Sham (indicated in the boxed regions). Thus, the increased accessibility at the *Sell* locus corresponds to increased transcription at that locus (*Figure 5j*) and a subsequent increase in the expression of CD62L (*Figure 4*). Importantly, the transcription of the additional phenotypic distinctions observed in *Figure 3* largely conformed wherein there was decreased expression of *Cx3cr1* and *Klrg1* (*Figure 5j*), though no change in expression was observed for other markers such *Cxcr3* and *Il7r*. Thus, sepsis leads to lasting changes in chromatin accessibility, some of which are concordant with gene expression. The resulting transcriptional changes are likely to reflect functional outcomes consistent with the composition of the memory CD8 T cell population.

To interrogate putative functional impairments, gene-set enrichment analysis (GSEA) was performed to compare Sham and CLP memory P14 CD8 T cells at D31 post-surgery. When evaluating the top five positively enriched KEGG pathways in CLP P14 CD8 T cells, there was an obvious trend toward cell cycling (*Figure 6a*). Specifically, the pathways included: KEGG_Ribosome,

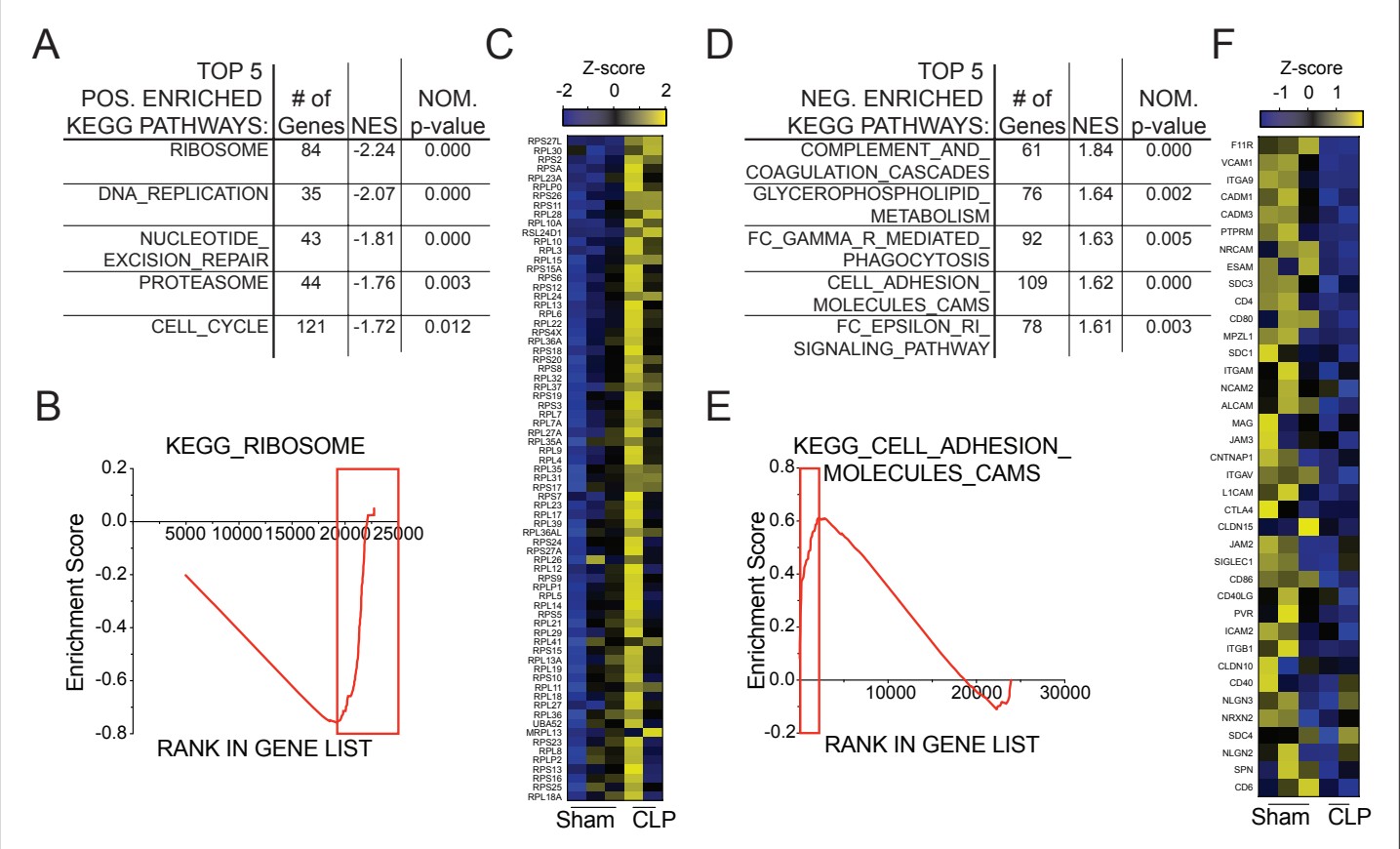

**Figure 6.** Gene set enrichment analysis (GSEA) reveals long-term sepsis-induced differences in molecular pathways of pre-existing memory CD8 T cells. Top 5 KEGG pathways positively- (**A**) and negatively- (**D**) enriched in CLP hosts. Enrichment scores for Ribosomal- (**B**) and Adhesion- (**E**) associated genes. Red box indicates leading edge of enriched region; genes enriched in CLP - box to right, genes enriched in Sham – box to left. Gene expression heatmap of core enriched genes for Ribosomal (**C**) and Adhesion (**F**) associated genes.

The online version of this article includes the following figure supplement(s) for figure 6:

**Source data 1.** Source data for **Figure 6**.

**Figure supplement 1.** Gene set enrichment analysis (GSEA) reinforces that sepsis promotes a shift to T$_{CM}$ at transcriptional level.

**Figure supplement 1—source data 1.** Source data for **Figure 6—figure supplement 1A-D**.

_DNA_Replication, _Nucleotide_Excision_Repair, _Proteasome, and _Cell_Cycle. In conjunction with the concordant gene analysis (**Figure 5g**), this information further supports the notion that the numerical recovery after sepsis alters the composition of memory CD8 T cells through proliferation. A critical example of this is the substantial enrichment of ribosomal proteins (**Figure 6b and c**), putatively necessary for increased translational output during proliferation. Conversely, several pathways were also negatively enriched in CLP P14 CD8 T cells (positively enriched in Sham), including: KEGG_Complement_And_Coagulation_Cascades, _Glycerophopholipid_Metabolism, _FC_Gamma_R_Mediated_Phagocytosis, _Cell_Adhesion_Molecules_CAMS, and _FC_Epsilon_RI_ Signaling_Pathway (**Figure 6d**). The primary underlying connection between the first two and last two of these appears to be linked to integrin expression and cell adhesion, while the change in glycerophospholipid metabolism may suggest sepsis-induced metabolic alterations. Given that integrin expression was also identified among the concordant genes in **Figure 5g**, gene enrichment in KEGG_Cell_Adhesion_Molecules_CAMS was evaluated and reduced expression of additional integrins was observed (**Figure 6e and f**). Given the critical nature of integrins in TCR function, including TCR-dependent function and immunologic synapse formation, these data suggest that sepsis alters the intrinsic capacity of pre-existing memory CD8 T cells to recognize cognate antigen. Further, when we compared the transcriptional changes between Sham D31 and CLP D31 with the published data

set KAECH_DAY15_EFF_VS_MEMORY_CD8_TCELL we observed that P14s from Sham hosts were biased toward effector CD8 T cells while the P14s from CLP hosts were biased toward memory CD8 T cells, mirroring the shift from effector to central CD8 T cell memory (*Figure 6—figure supplement 1*). This reinforces our observation of sepsis accelerating the adoption of time-dependent changes in the composition of the memory CD8 T cell pool.

## Sepsis-induced changes in pre-existing memory CD8 T cell composition impact cell function and capacity to control infection

To address the putative functional alterations resulting from sepsis-induced changes in the memory CD8 T cell pool, the capacity of memory P14 CD8 T cells to undergo TCR-dependent adhesion and immunologic synapse formation >30 days after either Sham or CLP surgery was assessed (*Figure 7—figure supplement 1a*). Notably, impairment in adherence capacity was observed in P14 CD8 T cells from CLP hosts under limiting stimulation conditions (low αCD3 concentration; *Figure 7—figure supplement 1b, c*); however, when stimulation was not limiting (high αCD3 concentration) Sham and CLP P14 CD8 T cells were equally capable of undergoing TCR-dependent adhesion. The TCR-induced signaling complex was then assessed via TIRF microscopy, under equivalent adherence conditions (high aCD3 concentration), to assess clustering of AKT, a surrogate of the TCR-induced signaling complex. Importantly, despite equal capability to adhere there remained a deficit in the ability to cluster AKT at the cell membrane following TCR stimulation (*Figure 7—figure supplement 1d, e*). Thus, sepsis leads to lasting changes in TCR based function of pre-existing memory CD8 T cells.

To address how these changes in signaling capability may influence cytokine production Sham and CLP splenocytes were disparately CFSE labeled >30 days post-surgery and then mixed for in vitro peptide stimulation (*Figure 7a*). Given that APCs from Sham and CLP hosts are shared in this scenario, discrepancies in function are not the result of differences in antigen display. Intriguingly, there was no deficiency in capacity to produce IFNγ, yet P14 CD8 T cells from CLP hosts actually had a higher capacity to produce IL-2 (*Figure 7b–d*). Importantly, similar results were observed after peptide (GP33) stimulation of the endogenous virus-specific memory CD8 T cell population (*Figure 7—figure supplement 2*). Indeed, this finding conforms precisely with the shift toward $T_{CM}$ in CLP hosts since $T_{CM}$ have greater capacity to produce IL-2 than $T_{EM}$. These data also suggest that changes in the composition of pre-existing memory CD8 cells may dominantly impact the function of the population as a whole.

With the relationship between the composition of the memory CD8 T cell pool and their capacity to promote effector function in mind, we next interrogated the ability of these memory CD8 T cells to control infection. Nolz et al. previously demonstrated that $T_{EM}$ more effectively control virulent *Listeria monocytogenes* (*L.m.*) infection compared to $T_{CM}$ (*Nolz and Harty, 2011*), likely due to localization of cells in either non-lymphoid tissues (critically the liver) or lymphoid tissues, respectively. Therefore, to address whether the shift toward $T_{CM}$ in the CLP host impaired the subsequent capacity to control *L.m.*, P14 CD8 T cells were enriched from either Sham or CLP hosts > 30 days post-surgery then transferred to naïve recipients. Transfer of these cells into naive recipients alleviates potential environmental deficits imposed by sepsis and allows for direct assessment of the capacity of the memory CD8 T cells to control infection. Additionally, the use of naive recipients alleviates confounding variables such as bystander responses (*Ehl et al., 1997*; *Lertmemongkolchai et al., 2001*). One day after transfer, mice that received either no cell transfer, P14 CD8 T cells from Sham mice, or P14 CD8 T cells from CLP were challenged with virulent *L.m.* expressing $GP_{33}$. *L.m.* challenge occurred 1 day after cell transfer to allow time for the cells to distribute and localize to their respective niches. $GP_{33}$ expression by *L.m.* enables the memory P14 CD8 T cells to mediate antigen-specific control. Colony-forming units (CFU) in both the liver and spleen were assessed 5 days post-infection (*Figure 7e*). Recipients that received memory P14 CD8 T cells from either Sham or CLP hosts more robustly controlled *L.m.* infection than the naive hosts that did not receive any memory CD8 T cells (*Figure 7f and g*). However, memory P14 CD8 T cells from Sham recipients were significantly better at controlling *L.m.* than those from CLP recipients; 77- and 20-fold differences control in the liver and spleen, respectively (*Figure 7f and g*). This improved control by memory P14 CD8 T cells from Sham hosts demonstrates the higher capacity of $T_{EM}$ to control *L.m.* infection. Thus, our data cumulatively demonstrate how sepsis-induced changes in the composition of the pre-existing memory CD8 T cells alters the functional capability of the memory CD8 T cell population as a whole, thereby altering the host response to infection.

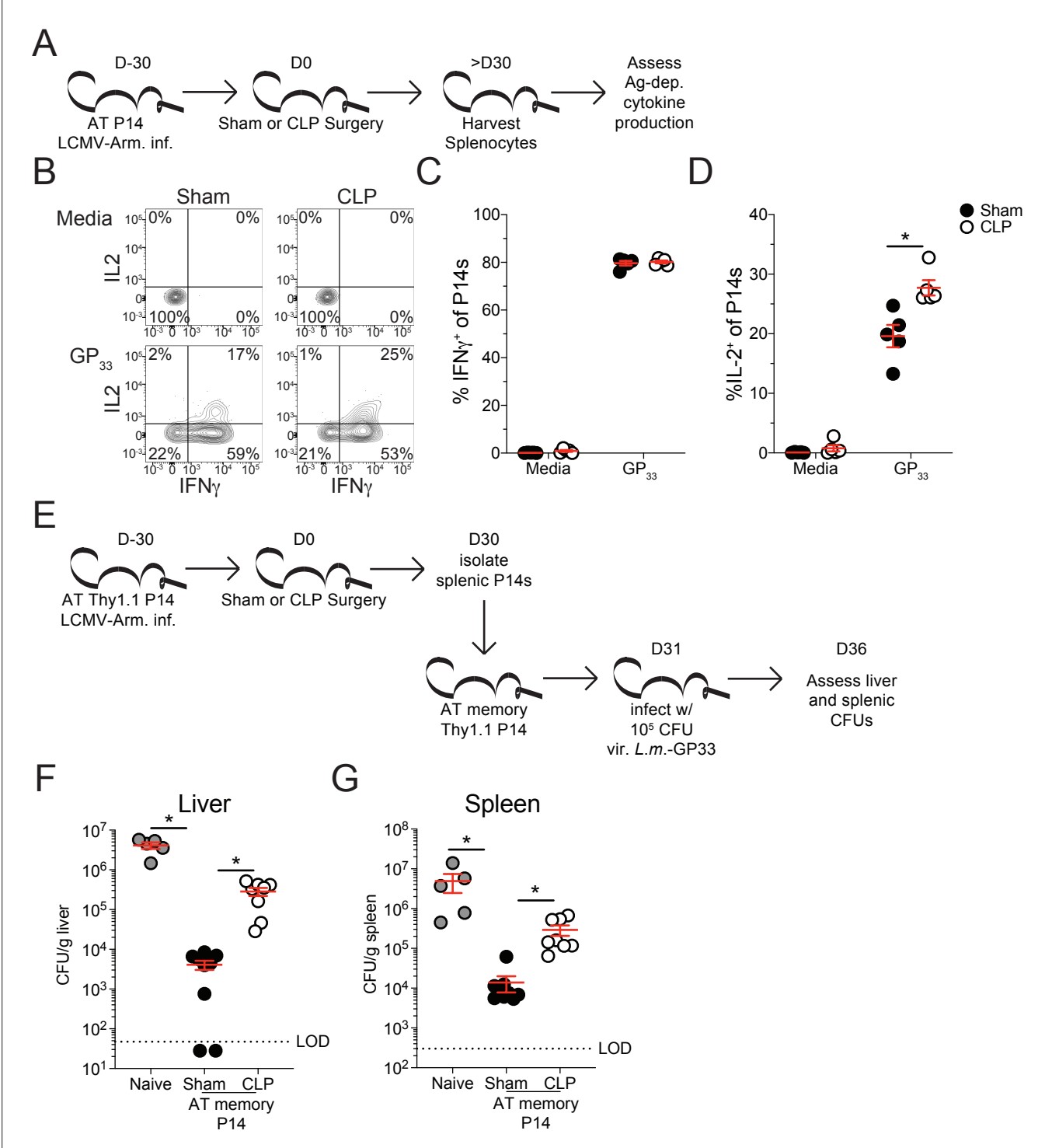

**Figure 7.** Sepsis leads to lasting changes in pre-existing memory CD8 T cell function and *Listeria* control. (**A**) Experimental Design: Antigen-experienced P14 chimeric mice were generated by adoptive transfer of 5 × 10³ naive Thy1.1⁺ TCR-transgenic P14 CD8 T cells to Thy1.2⁺ C57Bl/6 mice that were subsequently infected with LCMV-Arm. Mice underwent Sham or CLP surgery 30 days after infection. Splenocytes from Sham and CLP mice were isolated 30 days after surgery and disparately labeled with CFSE, mixed, and then placed in media alone (i.e. unstimulated) or stimulated GP₃₃ peptide. Representative profiles (**B**) and quantification of the frequency of IFN γ - (**C**) and IL-2- (**D**) producing P14s stimulated with either media control or GP₃₃. Data are representative of two independent experiments with 5 mice per group. (**E**) Experimental Design: Antigen-experienced P14 chimeric mice were generated by adoptive transfer of 5 × 10³ naive Thy1.1⁺ TCR-transgenic P14 CD8 T cells to Thy1.2⁺ C57Bl/6 mice that were subsequently infected with LCMV-Arm. Mice underwent Sham or CLP surgery 30 days after infection. Splenic P14 CD8 T cells were enriched from Sham and CLP mice

*Figure 7 continued on next page*

*Figure 7 continued*

30 days after surgery and then transferred into naïve mice. Mice that received either Sham or CLP P14 CD8 T cells, or did not receive any cell transfer (i.e. naïve) were then infected with $10^5$ CFU of *Listeria monocytogenes* expressing $GP_{33}$ (*L.m.*-$GP_{33}$) 1 day later. CFU of *L.m.*-$GP_{33}$ per gram of liver (**F**) and spleen (**G**) was assessed 5 days after infection. Data are cumulative of two independent experiments with 5–9 mice per group. *=p < 0.05. Error bars indicate standard error of the mean.

The online version of this article includes the following figure supplement(s) for figure 7:

**Source data 1.** Source data for *Figure 7C and D*.

**Source data 2.** Source data for *Figure 7F and G*.

**Figure supplement 1.** Sepsis leads to lasting deficit in pre-existing memory CD8 T cell TCR-dependent adhesion and immunologic synapse formation.

**Figure supplement 1—source data 1.** Source data for *Figure 7—figure supplement 1c*.

**Figure supplement 1—source data 2.** Source data for *Figure 7—figure supplement 1e*.

**Figure supplement 2.** Sepsis leads to lasting changes in pre-existing polyclonal memory CD8 T cell function.

**Figure supplement 2—source data 1.** Source data for *Figure 7—figure supplement 1*.

## Discussion

In the present study, we demonstrate that sepsis leads to a lasting change in the composition of the memory CD8 T cell compartment in the sepsis survivors. This occurs as a result of proliferation in the lymphopenic environment that occurs after sepsis, seen in both patients and mice, wherein $T_{CM}$ have higher proliferative capacity than $T_{EM}$. This biasing toward $T_{CM}$ alters the localization of memory CD8 T cells. Further, the memory CD8 T cell pool has an altered transcriptional landscape and chromatin accessibility, which is associated both with the transition toward $T_{CM}$ and functional alterations. The culmination of these sepsis-induced changes alters the function of the memory CD8 T cells and reduces their capacity to control virulent *L.m.* infection.

There are several important implications of the present study, and the biasing toward $T_{CM}$, that are relevant to our understanding of the immunoparalysis state. Among these is the relationship to tissue resident memory CD8 T cells ($T_{RM}$), which provide sensing and alarm function at sites of prior infection (*Masopust et al., 2001*; *Schenkel et al., 2013*). The present study focuses on the influence of sepsis on circulating $T_{EM}$ and $T_{CM}$ CD8 T cells; however, the substantial population of $T_{RM}$ throughout the body may be an interesting source of future interrogation. Danahy et al. previously demonstrated that $T_{RM}$ were not susceptible to sepsis-induced lymphopenia, due to their exclusion from the vasculature (*Danahy et al., 2017*). In the present study, recovery in cellularity was observed with time after sepsis, but the biasing toward $T_{CM}$ may pose a particular problem for $T_{RM}$. Specifically, Slütter et al. demonstrated that lung $T_{RM}$ are seeded from circulating $T_{EM}$ (*Slütter et al., 2017*). Thus, the challenge to the $T_{RM}$ may be twofold: (1) the reduced seeding of cells during the lymphopenic state and (2) reduced $T_{EM}$ pool from which to seed the $T_{RM}$. This reduction may culminate in a more rapid waning of lung $T_{RM}$ and reinforce susceptibility to previously encountered infections. Moreover, the detrimental effects of sepsis on memory CD8 T cells may also be relevant to other major inflammatory events and poorly controlled infections (e.g. SARS-CoV-2) and should be considerations in the long-term consequences for similarly impacted individuals (*Li et al., 2020*; *Sariol and Perlman, 2020*).

While the immunoparalysis state is often viewed directly through the lens of the detriments that may arise, it is also relevant to consider other means by which the host immune response may be shaped. The loss of $T_{EM}$ here demonstrates their critical role in fighting some infections (i.e. *L.m.*) (*Nolz and Harty, 2011*); however, $T_{CM}$ are also a potent population that can critically mediate control in other infection scenarios. Thus, infections for which $T_{CM}$ can provide critical control may be unimpaired or even enhanced in the post-septic environment. This is potentially complicated by intrinsic deficits that may be present in the memory cells, as observed in our GSEA analysis and by CD8 T cell extrinsic impairments. Therefore, future studies should consider additional interrogation of mechanisms by which the immune system is altered beyond detrimental aspects. Additionally, while our interrogation focused on a shift of pre-existing memory CD8 T cells toward a central memory phenotype, it remains possible that other T cell populations (e.g. different antigen specificities) may bias toward effector memory. This may be particularly relevant to memory T cell populations whose TCR has some low degree of cross reactivity with antigens present on microbes released during the septic event. These

considerations may be important for future therapeutic interrogation in the specific targeting of the appropriate deficits.

Additionally, it is relevant to consider that proliferation, as demarcated by Ki67, was also observed in naïve CD8 T cells of septic patients. While not the focus of the present study this is also observed in our mouse model. The proliferation by naïve CD8 T cells in septic hosts suggests that CD8 T cells are proliferating in response to the lymphopenic environment. Indeed, naive cells undergo antigen-independent proliferation in other lymphopenic environments, such as $Rag^{-/-}$ or irradiated hosts, wherein they adopt conventional markers of antigen experience along with some effector function-ality (*Cheung et al., 2009*; *Pribikova et al., 2018*; *Unsinger et al., 2009*; *White et al., 2017*). Thus, it may be relevant to consider how the proliferation of these cells also alter the composition of the memory CD8 T cell compartment and shapes host response to subsequent infection for which they may be specific.

Our novel characterization of how numeric recovery in the lymphopenic environment alters the composition of the memory CD8 T cell compartment demonstrates how sepsis can lead to lasting changes in host immunity. However, the implications of these changes may extend beyond the enhanced susceptibility to infection described here to potentially reframe our understanding of the immunoparalysis state. Future interrogation of these lasting effects will likely be required to best address the deficits that arise in the immunoparalysis state. Further understanding how sepsis shapes both naive and memory T cells may also alternately produce therapeutic interventions to benefit other diseases. One such example may be in the promotion of $T_{CM}$ over $T_{EM}$, or vice versa, for specific vaccination strategies. Such outcomes and lines of investigation would be highly instructive for under-standing how prior immune history shapes subsequent host immune responses.

# Materials and methods

## Key resources table

| Reagent type (species) or resource | Designation | Source or reference | Identifiers | Additional information |
|---|---|---|---|---|
| Strain, strain background (*Mus musculus*) | C57BL6/J | Jackson Laboratory | Stock No: 000664 (RRID:IMSR_ JAX:000664) | |
| Strain, strain background (*Mus musculus*) | B6.PL(84NS)/Cy | Jackson Laboratory | Stock No: 000983 (RRID:IMSR_ JAX:000406) | C57BL6/J Thy1.1 |
| Strain, strain background (*Mus musculus*) | B6.Cg-Tcratm1Mom Tg(TcrLCMV)327Sdz (P14) | Jackson Laboratory | Stock No: 37394-JAX (RRID:IMSR_ TAC:4138) | |
| Strain, strain background (*Mus musculus*) | Thy1.1/1.1- B6.Cg-Tcratm1Mom Tg(TcrLCMV)327Sdz | This paper | Thy1.1/1.1 P14 | Can be acquired through lab contact or breeding of above commercially available strains |
| Strain, strain background (Lymphocytic choriomeningitis virus) | Lymphocytic choriomeningitis virus Armstrong strain (LCMV-Arm) | Armstrong, C. and Lillie, R.D. Experimental lymphocytic choriomeningitis of monkeys and mice produced by a virus encountered in studies of the 1933 St Louis encephalitis epidemic. *Public Health Reports* 49, 1019–1027 (1934) | LCMV-Arm | Can be acquired through lab contact. |
| Strain, strain background (virulent Listeria monocytogenes) | Virulent recombinant *Listeria monocytogenes* expressing GP33-41 (XFL203 *L.m.*-GP33) | Shen et al. Recombinant Listeria monocytogenes as a live vaccine vehicle for the induction of protective anti-viral cell-mediated immunity. *PNAS* 92(9) 3987–3991 (1995) | *L.m.*-GP33 | Can be acquired through lab contact. |
| Peptide, recombinant protein | GP33-44 | AnaSpec | Catalog #: AS-61296 | |
| Antibody | CD8a (Rat monoclonal) | Biolegend | 5H10-1 (RRID:AB_312762) | FACs (1:400) |
| Antibody | CD11a (Rat monoclonal) | Biolegend | M17/4 (RRID:AB_312776) | FACs (1:300) |
| Antibody | Thy1.1 (Mouse monoclonal) | eBioscience | HIS51 (RRID:AB_1257173) | FACs (1:1000) |
| Antibody | KLRG1 (Mouse monoclonal) | eBioscience | 2F1 (RRID:AB_540279) | FACs (1:100) |

*Continued on next page*

*Continued*

| Reagent type (species) or resource | Designation | Source or reference | Identifiers | Additional information |
|---|---|---|---|---|
| Antibody | CD127 (Rat monoclonal) | eBioscience | eBioSB/199 | FACs (1:100) |
| Antibody | CD62L (Rat monoclonal) | Biolegend | MEL-14 (RRID:AB_1853103) | FACs (1:100) |
| Antibody | CX3CR1 (Mouse monoclonal) | eBioscience | SA011F11 (RRID:AB_2565701) | FACs (1:100) |
| Antibody | CXCR3 (Armenian Hamster monoclonal) | eBioscience | CXCR3-173 (RRID:AB_1210593) | FACs (1:100) |
| Antibody | CD27 (Armenian Hamster monoclonal) | eBioscience | LG.7F9 | FACs (1:100) |
| Antibody | CD69 (Hamster monoclonal) | Biolegend | H1.2F3 (RRID:AB_1853105) | FACs (1:100) |
| Antibody | CD103 (Hamster monoclonal) | Biolegend | 2E7 (RRID:AB_469040) | FACs (1:100) |
| Antibody | CD25 (Mouse monoclonal) | Biolegend | PC61.5 | FACs (1:100) |
| Antibody | CD122 (Rat monoclonal) | Biolegend | TM-b1 | FACs (1:100) |
| Antibody | IFNγ (Rat monoclonal) | eBioscience | XMG1.2 (RRID:AB_465410) | FACs (1:100) |
| Antibody | IL-2 (Rat monoclonal) | Biolegend | JES6-5H4 (RRID:AB_315298) | FACs (1:100) |
| Antibody | Ki67 (Mouse monoclonal) | BD Pharmingen | B56 (RRID:AB_2858243) | FACs (1:100) |
| Antibody | BrdU (Mouse monoclonal) | Biolegend | Bu20a (RRID:AB_1595472) | FACs (1:100) |
| Antibody | CD45RA (Mouse monoclonal) | Tonbo | HI100 | FACs (1:100) |
| Antibody | CD45RO (Mouse monoclonal) | Tonbo | UCHL1 | FACs (1:100) |
| Antibody | CD3 (Mouse monoclonal) | Biolegend | HIT3a | FACs (1:100) |
| Antibody | CD8a (Mouse monoclonal) | Biolegend | HIT8a | FACs (1:100) |
| Antibody | CCR7 (Mouse monoclonal) | Biolegend | G043H7 | FACs (1:100) |
| Antibody | CD3 (Mouse monoclonal) | Biolegend | OKT3 | Plate coating (0–10 µg) |
| Antibody | CD8a (Rat monoclonal) | Biolegend | 53–6.7 | FACs (1:100) |
| Antibody | AKT (rabbit monoclonal) | Cell Signaling Technology | 11E7 | TIRF microscopy (1:20) |
| Antibody | Anti-rabbit IgG (donkey monoclonal) | Biolegend | Poly4064 | TIRF microscopy (1:100) |
| Commercial assay or kit | Foxp3/ Transcription Factor Staining Buffer Set | Invitrogen | 00-5523-00 | |
| Software, algorithm | GraphPad Prism | GraphPad Prism 8 | Version 8.4.2 (464) (RRID:SCR_002798) | |

## Mice

Inbred C57Bl/6 (B6, Thy1.2) and TCR-transgenic (TCR-Tg) P14 (Thy1.1) mice were purchased from the National Cancer Institute (Frederick, MD) and maintained in the animal facilities at the University of Iowa at the appropriate biosafety level according to the University of Iowa Animal Care and Use Committee and National Institutes of Health guidelines. Male and female mice > 6 weeks of age were used for experiments; no discernable differences were observed based on sex of the animals.

## Generation of antigen-experienced CD8 T cells; P14 chimeras

To generate antigen-experienced CD8 T cells $5 \times 10^3$ naïve P14 TCR-Tg CD8 T cells were adoptively transferred into recipient mice, followed a day later by infection with $10^5$ plaque forming units (PFU) of Lymphocytic Choriomeningitis Virus-Armstrong (LCMV-Arm) by intraperitoneal (i.p.) injection.

## Institutional setting and IRB approval

Patients were recruited at the University of Iowa Hospitals and Clinics, an 811-bed academic tertiary care center. Blood sample acquisition, patient data collection, and analysis were approved by the University of Iowa Institutional Review Board (ID #201804822). Informed consent was obtained from patients or their legally authorized representatives.

## Sepsis patient selection and data collection

Subjects 18 years of age or older meeting Sepsis-3 criteria for sepsis or septic shock (*Singer et al., 2016*) secondary to intra-abdominal infection, soft tissue infection, bloodstream infection, or pneumonia were enrolled. Exclusion criteria were infection requiring antibiotics in the past month, hospitalization for infection in the past year, and chemotherapy or radiation within the past year were excluded. Demographics and baseline characteristics including age, gender, race, APACHE II score, SOFA score, and presence of septic shock were collected. EDTA-treated blood samples were collected within 24 hr of presentation.

## Healthy control patient selection and data collection

Healthy volunteers 25–80 years of age were recruited from University of Iowa faculty, staff, and graduate/professional students. Exclusion criteria were signs or symptoms of active infections, infection requiring antibiotics within the past month, infection requiring hospitalization in the past year, and chemotherapy or radiation in the past year. Demographic data including age, gender, and race were collected. EDTA-treated blood samples were collected at an initial visit to our research clinic.

## Human cell isolation and cryopreservation

Human cell isolation was adjusted from the previously described methodology (*Lauer et al., 2017*). Briefly, whole blood was centrifuged, and plasma removed. ACK red blood cell lysis buffer was then added to the cell pellet and rested for 5 min at room temperature. Cells were again centrifuged, and supernatant was removed. Lysis and centrifugation was repeated one to two additional times. Cells were then washed with PBS three times before being counted and resuspended in cell freeze media (90%FCS [Hyclone] 10%DMSO [Fischer Scientific]). Cells were then stored at –80 °C until use. When used in vitro, PBL were rapidly thawed and placed into warmed complete media. Cells were then washed three times with warmed media and aggregates filtered prior to use.

## Cell isolation

Peripheral blood was collected by submandibular cheek bleeds to obtain PBL. Single-cell suspensions from spleen, liver, and lymph nodes were generated after mashing tissue through a 70 µm cell strainer without enzymatic digestion. Liver cells were subsequently run on a 35 % Percoll gradient. ACK lysis buffer was used for red blood cell lysis of PBL, spleen, and liver samples.

## Flow cytometry, peptides, and cytokine detection

Flow cytometry data were acquired on a FACSCanto or LSRII (BD Biosciences, San Diego, CA) and analyzed with FlowJo software (Tree Star, Ashland, OR). FlowJo Software was also used for FlowSOM and tSNE analysis. To determine expression of cell surface proteins, mAb were incubated at 4 °C for 20–30 min and cells were fixed using Cytofix/Cytoperm Solution (BD Biosciences) and, in some instances, followed by incubation with mAb for an additional 20–30 min to detect intracellular proteins. The following mAb clones were used to stain murine samples: CD8a (53–6.7; eBioscience), CD11a (M17/4; Biolegend), Thy1.1 (HIS51; eBioscience), KLRG1 (2F1; Biolegend), CD127 (eBioSB/199; eBioscience), CD62L (MEL-14; eBioscience), CX3CR1 (SA011F11; Biolegend), CXCR3 (CXCR3-173; Biolegend), CD27 (LG.7F9; eBioscience), CD69 (H1.2F3; Biolegend), CD103 (2E7; eBioscience), CD25 (PC61.5; eBioscience), CD122 (TM-b1; eBioscience), IFNγ (XMG1.2; eBioscience), IL-2 (JES6-5H4; eBioscience), Ki67 (B56; eBioscience) and BrdU (Bu20a; eBioscience). The following mAb clones were

used staining of patient samples: CD45RA (HI100; Tonbo), CD45RO (UCHL1; Tonbo), CD3 (HIT3a; Biolegend), CD8a (HIT8a; Biolegend), and CCR7 (G043H7; Biolegend). Overnight fixation with FoxP3 fixation/permeabilization (eBioscience) buffer was used to stain Ki67 and BrdU. For BrdU staining, following fixation/ permeabilization cells were treated with DNAse I for 1 hr at 37 °C, then stained for intracellular BrdU.

## Cecal ligation and puncture (CLP) model of sepsis induction

Mice were anesthetized with ketamine/xylazine (University of Iowa, Office of Animal Resources), the abdomen was shaved and disinfected with Betadine (Purdue Products), and a midline incision was made (*Sjaastad et al., 2020a*). The distal third of the cecum was ligated with Perma-Hand Silk (Ethicon), punctured once (for $CLP_{20}$) or twice (for $CLP_{50}$) using a 25-gauge needle, and a small amount of fecal matter extruded out of each puncture. The cecum was then returned to abdomen, the peritoneum was closed with 641 G Perma-Hand Silk (Ethicon), and skin sealed using surgical Vetbond (3 M). Following surgery, 1 mL PBS was administered s.c. to provide post-surgery fluid resuscitation. Bupivacaine (Hospira) was administered at the incision site, and flunixin meglumine (Phoenix) was administered for postoperative analgesia. Sham mice underwent identical surgery excluding cecal ligation and puncture.

## Normalized assessment of lymphopenia (Figure 2C)

Due to large differences in the number of naïve to antigen-experienced endogenous cells to antigen-experienced P14 CD8 T cells in order to compare the relative degree of lymphopenia the data for each population was normalized. Data are normalized as: % survival = (1-((# of [Naive, Endo, or P14] CD8 T cells in the PBL of the same mouse prior to surgery) - (# of [Naive, Endo, or P14] CD8 T cells in the PBL of a mouse at D2 post-surgery)) / (# of [Naive, Endo, or P14] CD8 T cells in the PBL of the same mouse prior to surgery)) * 100.

## BrdU administration

BrdU was administered by a single i.p. injection (2 mg/mouse) followed by ad libitum consumption in the drinking water (0.8 mg/mL) for 7 days.

## RNA-seq and gene set enrichment analysis

Total RNA was extracted from P14 (Thy1.1$^+$CD8a$^{lo}$CD11a$^{hi}$) CD8 T cells sorted 1 day post-Sham or CLP and 31 days post-Sham or CLP, 2–3 biological replicates were obtained for each group. Libraries were sequenced on Illumina's HiSeq2000 in single-end mode with the read length of 50 nucleotides. The RNA-seq data are deposited at the GEO (GSE174358) under the SuperSeries of GSE174359. RNA-seq was performed as previously described (*Shan et al., 2017*). The sequencing quality of RNA-seq libraries was assessed by FastQC v0.11.4 (http://www.bioinformatics.babraham.ac.uk/projects/fastqc/). Adaptor sequences were removed through Cutadapt. The reads were mapped to mouse genome mm9 using Tophat (v2.1.0) (*Trapnell et al., 2009*). Mapped reads were then processed by Cuffdiff (v2.2.1) to estimate the expression level of all genes and identify differentially expressed genes. The expression level of a gene was expressed as a gene-level Fragments Per Kilobase of transcripts per Million mapped reads (FPKM) value. Upregulated or downregulated genes in when comparing groups were identified by requiring a greater than 1.5-fold expression change and a false discovery rate (FDR) < 0.1, as well as a FPKM values > 1.0. The reproducibility of RNA-seq data was evaluated by applying the principal component analysis for all genes between biological replicates. UCSC genes from the iGenome mouse mm9 assembly (https://support.illumina.com/sequencing/sequencing_software/igenome.html) were used for gene annotation. Gene set enrichment and functional assignment were performed in software from the Broad Institute as described (*Martin et al., 2015*; *Shan et al., 2017*; *Subramanian et al., 2005*). Enrichment was evaluated for Day 31 CLP samples relative to Day 31 Sham samples.

## ATAC-seq and data analysis

To determine the global impact of sepsis on chromatin accessibility, splenic memory P14 CD8 T cells were sorted from Sham and CLP hosts > 30 days after surgery. $5 \times 10^4$ cells were prepared for sequencing as previously described (*Buenrostro et al., 2015*; *Shan et al., 2021*). The ATAC-seq data

are deposited at the GEO (accession number GSE174357) under the SuperSeries of GSE174359. The sequencing quality of ATAC-seq libraries was assessed by FastQC v0.11.4 (http://www.bioinformatics. babraham.ac.uk/projects/fastqc/) and adapters were removed through Cutadapt. The reads were mapped to mouse genome mm9 using Bowtie2 v2.2.5 and only uniquely mapped reads (MAPQ >10) were retained. The mapped reads from multiple replicates were pooled for Sham or CLP CD8$^+$ T cells, respectively, and were processed with MACS v2.1. 1 (*Zhang et al., 2008*) for peaks calling, with stringent criteria of ≥4 fold enrichment, $P$-value < 1E–five and FDR < 0.05. These sites were merged to generate a union pool of chromatin accessible sites containing 43,784 unique sites. For reproducibility analysis, reads at each site were counted in each ATAC-seq library, and then normalized by the total read-count of the union sites in the respective library. The resulting matrix was used for the principal component analysis. The read count matrix was used as input for edgeR (*Robinson et al., 2010*) (v.3.20.7.2) (quasi-likelihood test, robust, fold-change 2.0 and FDR < 0.01) to identify differential chromatin accessible sites between P14_Sham and P14_CLP conditions. A total of 304 Sham-specific and 1342 CLP-specific sites were identified, respectively.

## CFSE

Splenocytes ($10^7$ /mL) from CLP and Sham hosts were labeled with CarboxyFluorescein diacetate Succinimidyl Ester (CFSE; eBioscience) by incubating the cells at room temperature for 15 minutes with 1 µM or 0.1 µM CFSE, respectively. Labeled cells were then incubated for 5 minutes with 1 mL FCS on ice to remove any free CFSE, and washed three times with RPMI prior to stimulation.

## Peptide stimulation

CFSE labeled splenocytes from Sham and CLP hosts were mixed 1:1 and stimulated with 200 nM of GP$_{33}$ peptide or media control for 8 hr at 37 °C in the presence of Brefeldin A (BfA; BD Biosciences).

## Listeria challenge

Memory P14 CD8 T cells were isolated from either Sham or CLP hosts by positive selection, based on Thy1.1 expression, and naïve recipients received 2 × $10^5$ of either P14 CD8 T cells each (controls did not receive cell transfer). Mice were subsequently infected the following day with $10^5$ colony forming units (CFU) of virulent *Listeria monocytogenes* (10,403 s) express the GP$_{33}$ epitope (*L.m.*-GP33).

## Adhesion assay

Cellular adhesion was performed as previously described with some modification (*Bilal et al., 2015*; *Chapman et al., 2012*). Briefly, flat-bottomed 96-well plates (Thermo-Fisher) were coated with 0–10 µg of αCD3 (OKT3, Biolegend). P14 CD8 T cells were isolated by positive selection, based on Thy1.1. 5 × $10^6$ P14 CD8 T cells were incubated on the plate for 30 min. Non-adherent cells were removed by quickly inverting the plate to empty contents. Adherent cells were stained with αCD8a-APC-Cy7 (53–6.7; Biolegend). Cells were washed twice with PBS before being imaged utilizing Licor Odyssey Infrared detector.

## TIRF microscopy

Images were taken using Leica AM TIRF MC imaging system as described with the following modifications (*Bilal et al., 2015*). P14 CD8 T cells were isolated by positive selection, based on Thy1.1, and placed on glass chamber slides (5 × $10^4$ cells/chamber; LabTek II) precoated with 10 µg/mL α-CD3 mAb. Cells were stimulated for 15 minutes, fixed with 4 % paraformaldehyde, and permeabilized with 0.25 % Triton-X. Cells were blocked with SEA blocking buffer (Thermo-Fisher) for 1 hour and stained with 5 µL rabbit α-human/mouse AKT antibody (11E7, Cell Signaling Technology) overnight at 4 °C. Cells were washed and incubated with DyLight 488-conjugated donkey α-rabbit IgG (poly4064, BioLegend) secondary antibody for 2 hr at room temperature. Cells were washed and fresh PBS was added to each well. Images were taken at room temperature using 100 X oil submersion lens and Leica AM TIRF MC imaging system at the University of Iowa Central Microscopy Research Facility. Laser intensity and exposure parameters remained constant within each experiment. TIRF microscopy images were analyzed using ImageJ software. Membrane AKT was quantified by measuring mean pixel intensity in the longest axis of cells.

## Statistical analysis

Unless stated otherwise data were analyzed using Prism8 software (GraphPad) using two-tailed Student t-test (for two individual groups, if unequal variance Mann-Whitney U test was used), one-way ANOVA with Bonferroni post-hoc test (for >2 individual groups, if unequal variance Kruskal-Wallis with Dunn's post-hoc test was used), two-way ANOVA (for multiparametric analysis of two or more individual groups, pairing was used for samples that came from the same animal), Fisher's exact test (for categorical data from two individual groups) with a confidence interval of >95% to determine significance (*p ≤ 0.05). Data are presented as standard error of the mean.

## Acknowledgements

We thank our labs and collaborators for their useful discussion. This work was supported by NIH Grants R01AI114543 (VPB), R21AI147064 (VPB), R35GM134880 (VPB), R21AI151183 (VPB), R01GM115462 (TSG), R35GM140881 (TSG), R01AI112579 (H-HX), R01AI121080 (H-HX and WP), R01AI139874 (H-HX and WP), R21AI157121 (JCH), T32AI007511 (IJJ), T32AI007485 (IJJ) and a Veterans Health Administration Merit Review Award I01B × 001324 (TSG). Research produced at the Central Microscopy Research Core at the University of Iowa was supported by the National Cancer Institute of the National Institutes of Health [P30CA086862].

## Additional information

### Funding

| Funder | Grant reference number | Author |
|--------|------------------------|--------|
| National Institutes of Health | R01AI114543 | Vladimir P Badovinac |
| National Institutes of Health | R21AI147064 | Vladimir P Badovinac |
| National Institutes of Health | R21AI151183 | Vladimir P Badovinac |
| National Institutes of Health | R01GM115462 | Thomas S Griffith |
| National Institutes of Health | R35GM134880 | Vladimir P Badovinac |
| National Institutes of Health | R35GM140881 | Thomas S Griffith |
| National Institutes of Health | R01AI112579 | Hai-Hui Xue |
| National Institutes of Health | R01AI121080 | Hai-Hui Xue Weiqun Peng |
| National Institutes of Health | R01AI139874 | Hai-Hui Xue Weiqun Peng |
| National Institutes of Health | R21AI157121 | Jon CD Houtman |
| National Institutes of Health | T32AI007511 | Isaac J jensen |
| National Institutes of Health | T32AI007485 | Isaac J jensen |
| Veterans Health Administration HSR and D | I0BX001324 | Thomas S Griffith |
| National Cancer Institute | P30CA086862 | Jon CD Houtman |

| Funder | Grant reference number | Author |
|--------|------------------------|--------|

The funders had no role in study design, data collection and interpretation, or the decision to submit the work for publication.

## Author contributions
Isaac J Jensen, Conceptualization, Data curation, Formal analysis, Investigation, Methodology, Writing – original draft, Writing – review and editing; Xiang Li, Patrick W McGonagill, Data curation, Formal analysis, Writing – review and editing; Qiang Shan, Data curation, Methodology, Writing – review and editing; Micaela G Fosdick, Data curation, Formal analysis, Visualization, Writing – review and editing; Mikaela M Tremblay, Methodology, Writing – review and editing; Jon CD Houtman, Resources, Writing – review and editing; Hai-Hui Xue, Weiqun Peng, Resources, Supervision, Writing – review and editing; Thomas S Griffith, Writing – review and editing; Vladimir P Badovinac, Conceptualization, Funding acquisition, Resources, Supervision, Writing – review and editing

## Author ORCIDs
Isaac J Jensen http://orcid.org/0000-0002-3107-3961
Micaela G Fosdick http://orcid.org/0000-0002-2427-532X
Hai-Hui Xue http://orcid.org/0000-0002-9163-7669
Thomas S Griffith http://orcid.org/0000-0002-7205-9859
Vladimir P Badovinac http://orcid.org/0000-0003-3180-2439

## Ethics
Human subjects: Patients were recruited at the University of Iowa Hospitals and Clinics, an 811-bed academic tertiary care center. Blood sample acquisition, patient data collection, and analysis were approved by the University of Iowa Institutional Review Board (ID #201804822). Informed consent was obtained from patients or their legally authorized representatives.

Experimental procedures using mice were approved by University of Iowa Animal Care and Use Committee under ACURF protocol #6121915 and #9101915. The experiments performed followed Office of Laboratory Animal Welfare guidelines and PHS Policy on Humane Care and Use of Laboratory Animals. Cervical dislocation was used as the euthanasia method of all experimental mice. Inbred C57Bl/6 (B6, Thy1.2) and TCR-transgenic (TCR-Tg) P14 (Thy1.1) mice were purchased from the National Cancer Institute (Frederick, MD) and maintained in the animal facilities at the University of Iowa at the appropriate biosafety level according to the University of Iowa Animal Care and Use Committee and National Institutes of Health guidelines. Male and female mice >6 weeks of age were used for experiments; no discernable differences were observed based on sex of the animals.

## Decision letter and Author response
Decision letter https://doi.org/10.7554/eLife.70989.sa1
Author response https://doi.org/10.7554/eLife.70989.sa2

# Additional files

## Supplementary files
• Transparent reporting form

## Data availability
Sequencing data are deposited in GEO under accession code GSE174358. Source data for all figures are provided in associated excel files.

The following dataset was generated:

| Author(s) | Year | Dataset title | Dataset URL | Database and Identifier |
|-----------|------|---------------|-------------|--------------------------|
| Peng B | 2021 | Sepsis leads to lasting changes in phenotype and function of memory CD8 T cells (RNA-Seq) | https://www.ncbi.nlm.nih.gov/geo/query/acc.cgi?acc=GSE174358 | NCBI Gene Expression Omnibus, GSE174358 |

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
