## [Decision Letter]

**Acceptance summary:**

This manuscript addresses the important issue of how sepsis influences the function particularly of CD8 T cells. It shows that while antigen-specific T cells respond to infection following transient sepsis and paralysis, embedded changes occur in the T cells influencing their long-term performance and capacity to mediate immune protection.

**Decision letter after peer review:**

Thank you for sending your article entitled "Sepsis leads to lasting changes in phenotype and function of memory CD8 T cells" for peer review at *eLife*. Your article is being evaluated by 2 peer reviewers, one of whom is a member of our Board of Reviewing Editors, and the evaluation is being overseen by Satyajit Rath as the Senior Editor. The following individual involved in review of your submission has agreed to reveal their identity: Antoine Roquilly (Reviewer #2).

A summary of the essential revisions required is provided below. The individual reviews from the reviewers are also provided for the sake of context.

Essential revisions:

1. The mice model used to discriminate endogen and transferred memory T cells is underexploited. While P14 T cells are LCMV-specific, they are not specific to the antigens produced during CLP surgery. The reasons to discriminate endogen (Thy 1.1neg) vs. P14 (Thy 1.1pos) during CLP are not clear. Most of the time, endogenous and P14 CD8 memory T cells have the same response (Figure 2C, 4D-E) while this information is missing and only P14 response is described in most of the experiments (Figure 3, Figure 4B-C, 4G, 5, 6 and 7). The comparison of endogenous CD8 memory T cells with P14 should be consistent throughout the study since the role of TCR signaling could be of importance (as suggested by the increased Cish gene which is involved in TCR functional inhibition, see RNAseq, Figure 5H). Finally, the in vivo functional assay of the memory T cells response (Figure 7E-G) does not exclude a bystander role of endogen cells since IL-2 production can act on both cell types.

2. The description by flow cytometry of different subsets of P14 cells during sepsis (Figure 3C-H) is of interest but the mechanisms explaining the increased in cluster 8 (CD62L+) is not clear. In addition, the definition of cell survival (Figure 2C) is not clear to me and should be better explained.

3. Clarification of RNA seq analyses: The authors endeavour to uncover a mechanism for the altered activity of the CD8 T cells and use bulk RNAseq. In this analysis they identify 3 clusters. It is not clear exactly how the 8 subsets identified by flow cytometry and the sequence clusters relate to each other. Cluster 3 is composed of 57 genes that are changed but none of the genes in this cluster were mentioned nor provided in a table for reviewer consideration.

The finding of a CM and EM signature in the transcriptomic analyses would have strengthened the results. Whether if this phenomena is common with endogen cells, and if this is associated with alteration of cell function is unknown. Notably, is the production of IL-2 during sepsis similar in CD62L+ and CD62L- cells?

Overall, the RNAsequencing analyses is difficult to interpret due to the highly curated presentation and lack of fuller datasets.

*Reviewer #1:*

In this work, the authors endeavoured to understand how CD8 T cells, and preexisting memory CD8 T cells, might be influenced by sepsis. This is a very important question. They uncover interestingly that memory cells can still respond to subsequent infections, but are indelibly modulated by their exposure to the septic environment. This work provides interesting insights to the modulation of T lymphocytes and subsequent responsiveness post sepsis.

This manuscript addresses the important issue of how sepsis influences the function particularly of CD8 T cells. It shows that while antigen-specific T cells respond to infection following transient sepsis and paralysis, embedded changes occur in the T cells influencing their long-term performance. Although the changes in T cells are interesting, these effects are not dissociated from the extrinsic influences of other cell types that might also be affected by sepsis.

Page 4, line 91/92: There appears to be a word or words missing.

More recent references eg. Roquilly et al., Immunity and Nature Immunology have not been referenced.

In the human samples, a gap in the study is that admission may not correspond to the onset of sepsis. Delineation of a surrogate for the timing would be useful and some de-identified data to provide a temporal perspective of the onset of sepsis. One possibility may be the timing of lymphopenia. Part of the information is provided in Table 1.

The initial data is developed through characterisation of the proportions and numbers of different subsets of CD8 T cells. The hypotheses that the differential expansion/contraction of these subsets is based on our knowledge of normal responses. It is not entirely clear that this will be the case during sepsis. Using clustering based on known surface receptor expression, the authors designate 8 different subsets which they posit a pattern of temporal development. They then go on to endeavour to uncover a mechanism for the altered activity of the CD8 T cells and use bulk RNAseq. In this analysis they identify 3 clusters. It is not clear exactly how the 8 subsets identified by flow cytometry and the sequence clusters relate to each other. Cluster 3 is composed of 57 genes that are changed but none of the genes in this cluster were mentioned nor provided in a table for reviewer consideration. The RNAsequencing analyses is difficult to interpret due to the highly curated presentation and lack of fuller datasets.

Overall, this is an interesting study but the data are presented in quite a superficial manner limiting the impact of the work.

*Reviewer #2:*

Isaac J. Jensen et al., investigated in septic humans and in a mice of peritonitis the time course of the modifications of memory T cells during and after sepsis. They tracked in vivo the fate of antigen-specific memory T cells by using an elegant Antigen-specific T cells transfer whose proliferation is induced by a first viral infection. Once the virus-specific memory T cells are well settled, they induced an intraabdominal sepsis to investigate the modification of memory T cells during and after sepsis. They found in septic humans that while the percentages of CD8 T cells among lymphocytes remained unchanged, the rates of proliferating naïve and memory CD8 T cells were increased during sepsis. In the mice models, they observed that the survival of memory T cells decreased during sepsis, but their number rapidly returned to control values due to high rate of in vivo proliferation. Yet, the recovery of the number of memory T cells is associated with modifications of the proportions of effector vs. central memory T cells, and of the tissue localization. Transcriptomic and epigenetic analyses confirmed modifications of the CD8 T cells functional programming during sepsis, with upregulation of cell survival and proliferation functions. Finally, the authors demonstrated that the cytokine production of antigen-specific memory T cells is not decreased during sepsis, and IL-2 being even increased. in vivo, the gain of function of antigen-specific CD8 T cells (high proliferation, high IL-2 production) are not associated with higher control of viral load during reinfection. Altogether, the authors demonstrated that while memory CD8 T cells gain function during sepsis, it is associated with a poorer control of viral infection. These data add in an interesting way to the ongoing discussion on whether sepsis induced training of immunity (gained functions and increased resistance to infection) or tolerance / immunosuppression (loss of functions and increased susceptibility to infection).

The conclusions of this paper are mostly well supported by data, but some aspects of data analysis need to be clarified and extended.

1) The mice model used to discriminate endogen and transferred memory T cells is underexploited. While P14 T cells are LCMV-specific, they are not specific to the antigens produced during CLP surgery. So the reasons to discriminate endogen (Thy 1.1neg) vs. P14 (Thy 1.1pos) during CLP are not clear. Most of the time, endogen and P14 CD8 memory T cells have the same response (Figure 2C, 4D-E) while this information is missing and only P14 response is described in most of the experiments (Figure 3, Figure 4B-C, 4G, 5, 6 and 7). The comparison of endogen CD8 memory T cells with P14 should be consistent throughout the study since the role of TCR signaling could be of importance (as suggested by the increased Cish gene which is involved in TCR functional inhibition, see RNAseq, Figure 5H). Finally, the in vivo functional assay of the memory T cells response (Figure 7E-G) does not exclude a bystander role of endogen cells since IL-2 production can act on both cell types.

2) The description by flow cytometry of different subsets of P14 cells during sepsis (Figure 3C-H) is of interest but the mechanisms explaining the increased in cluster 8 (CD62L+) is not clear. Are the modifications in phenotype observed in Figure 3 explained by the transcriptomic activity of cells? The finding of a CM and EM signature in the transcriptomic analyses would have strengthened the results. Whether if this phenomena is common with endogen cells, and if this is associated with alteration of cell function is unknown. Notably, is the production of IL-2 during sepsis similar in CD62L+ and CD62L- cells?

As a summary, data are sounds, but the demonstration that the alterations are specific, or not, to any memory CD8 T cells subsets; and are antigen-specific or not, would have significantly increased the gain of knowledge.

In general, the paper is difficult to follow because the studied cells frequently between Figures: endo. vs P14, CD62L+ vs. CD62Lneg, then 14 alone.

1 – I would recommend to analyse endogen and P14 cells together throughout the manuscript. Indeed, I think that more than the endogen vs. transfer feature, P14 cells are likely not responding directly to CLP-derived antigens.

2 – The definition of cell survival (Figure 2C) is not clear to me and should be better explained.

3 – While the data of CD62L+ vs CD62L- subsets are of interest (Figure 3), this information is not exploited in the RNAseq and ATAC-seq analyses. The comparison with public data set of effector memory and central memory T cells would likely reinforced the message of differential composition of these subsets.

---

## [Author Response]

Essential revisions:1. The mice model used to discriminate endogen and transferred memory T cells is underexploited. While P14 T cells are LCMV-specific, they are not specific to the antigens produced during CLP surgery. The reasons to discriminate endogen (Thy 1.1neg) vs. P14 (Thy 1.1pos) during CLP are not clear.

We apologize for the lack of clarity. The true value of using the P14s in this system is because, as the reviewer indicates, they are not specific for the antigens evoked/released after sepsis and therefore their response can be truly delineated from a sepsis-induced ‘secondary’ antigen response.

Additionally, because novel effector CD8 T cell responses are anticipated/predicted during the septic event this would complicate sole analysis of all antigen experienced (effector and memory) CD8 T cells. Therefore, memory P14 CD8 T cells serve as a sentinel population to describe how sepsis influences those pre-existing memory CD8 T cells that exist prior to sepsis. After all, that is the question experimentally addressed in this submission.

Text clarifying and extended this point has been incorporated within the manuscript (line 172-177).

Most of the time, endogenous and P14 CD8 memory T cells have the same response (Figure 2C, 4D-E) while this information is missing and only P14 response is described in most of the experiments (Figure 3, Figure 4B-C, 4G, 5, 6 and 7). The comparison of endogenous CD8 memory T cells with P14 should be consistent throughout the study since the role of TCR signaling could be of importance (as suggested by the increased Cish gene which is involved in TCR functional inhibition, see RNAseq, Figure 5H).

The reason for the seemingly selective use of the P14s in the indicated figures is due to potential complicating factors of using the polyclonal CD8 T cell responses. For instance:

– With respect to Figure 3 the findings are subsequently evaluated for the polyclonal antigen experienced CD8 T cell population in Figure 4 E.

– With regard to Figures 5 and 6 these were done on the memory P14 CD8 T cells because of potential complications associated with using the polyclonal antigen experienced CD8 T cell population (e.g., novel CD8 T cell responses to antigens induced by sepsis and the potential for cross-reactivity between intestinal microflora with LCMV epitopes). This is particularly relevant to indicated TCR signaling which may be influenced by varying TCR signal strengths elicited due to novel microflora directed responses or potential secondary cross-reactive responses. We therefore contend that by focusing on memory P14 CD8 T cells wherein the TCR is a fixed variable this gives additional power to the analysis that would not be achievable in polyclonal populations. Text emphasizing this point has been now incorporated (line 318-321).

– With respect to Figure 7A-D we had continued to focus on memory P14 CD8 T cells due to convenience/clarity but are more than happy to include the data emphasizing GP33 stimulated cytokine (IFNγ and IL-2) production for the endogenous memory CD8 T cells, that showed a similar trend, as well (Please see new Figure 7—figure supplement 2 and line 384-386).

– Regarding 7E-G, however, transfer of endogenous cells would result in the same complications highlighted in the previous bullet point therefore it is our assertion that such assessments would be more confounding than illuminating to the impact of sepsis on pre-existing memory CD8 T cells.

Finally, the in vivo functional assay of the memory T cells response (Figure 7E-G) does not exclude a bystander role of endogen cells since IL-2 production can act on both cell types.

With regard to this point it is unclear what endogenous cells the reviewer is referring to. If the concern is with regard to endogenous cells generated either by LCMV or sepsis then we recognize that this could be a complicating factor and had therefore isolated the Thy1.1 memory P14 CD8 T cells prior to adoptive transfer. If the concern is with regard to potential bystander roles of the endogenous effector and/or memory CD8 T cells in the transfer recipients this seems unlikely given that the recipients are all naïve SPF mice that have neither experienced LCMV infection or undergone surgery. Thus, the mice are on equal standing with regard to the groups except with respect to which donor the memory P14s came from (if any) and would therefore reasonably rule out a bystander role for endogenous cells in either case. Text clarifying and extended this point has been incorporated within the manuscript (line 398-402).

2. The description by flow cytometry of different subsets of P14 cells during sepsis (Figure 3C-H) is of interest but the mechanisms explaining the increased in cluster 8 (CD62L+) is not clear. In addition, the definition of cell survival (Figure 2C) is not clear to me and should be better explained.

The mechanism, as we describe it, for the increased representation in Tcm (CD62L+) CD8 T cells is through the increased proliferative capacity of Tcm relative to Tem. Thus as the numeric recovery is achieved through homeostatic proliferation the Tcm gradually become overrepresented (relative to sham hosts). This mechanism is probed in Figure 4B-E of the manuscript. Additional text clarifying this point has been incorporated to reduce confusion (line 260-262).

We apologize for the lack of clarity in Figure 2C. This is calculated based off of the number of cells in peripheral blood (with respect to the indicated subset) at day 2 post-surgery divided by the average number of cells in the peripheral blood (with respect to the indicated subset) prior to surgery multiplied by 100 (to convert to a percentage). This transformation of the data is done to because there are highly different numbers in the different cell population (Naïve, endogenous, and P14 CD8 T cells) and is meant to demonstrate that all populations are equally susceptible to sepsis induced lymphopenia (in spite of differences in the cell number). Text clarifying this point is now included in the Materials and methods (line 1010-1016).

3. Clarification of RNA seq analyses: The authors endeavour to uncover a mechanism for the altered activity of the CD8 T cells and use bulk RNAseq. In this analysis they identify 3 clusters. It is not clear exactly how the 8 subsets identified by flow cytometry and the sequence clusters relate to each other.

Because of the differences in how these distinct analyses with discrete outputs are performed it is likely not best practice to overlay phenotypic clusters with changes in population RNA-seq. However, as stated in the text, cluster 1 demonstrates an enhancement of time-dependent changes in the CLP P14s. When considering how the memory CD8 T cell pool naturally changes overtime to have a reduced frequency of Tem and a higher frequency of Tcm this cluster reflects the Tem to Tcm conversion induced by sepsis. Thus, the changes in clusters 6 and 8 of the flow cytometry data likely are encompassed in this first cluster.

Additionally, we performed additional analyses (new Figure 6—figure supplement 1) and when we compared our gene expression changes with changes in a published data set (KAECH_DAY15_EFF_VS_MEMORY_CD8_TCELL), P14s from CLP hosts were more similar to memory CD8 T cells whereas P14s from Sham hosts had a more effector like signature further reflecting (and confirming) the shift toward central memory subset in CLP hosts. Text has been incorporated to discuss these additional analyses and better highlight this point (line 355-361).

Cluster 3 is composed of 57 genes that are changed but none of the genes in this cluster were mentioned nor provided in a table for reviewer consideration.

We apologize for this oversight and have included the requested information for all 3 gene clusters in Table II and the associated file “Figure 5-source data 2”.

The finding of a CM and EM signature in the transcriptomic analyses would have strengthened the results.

As indicated in the comment regarding the overlay of phenotypic clusters with RNA-seq clusters we feel that cluster 1 likely captures this transcription signature. Further, while the fold change in *Sell* (gene for CD62L) did not meet the 1.5X threshold change it did have a statistically significant change in expression (see Figure 5J) which further supports this conversion. Notably the fold change in the gene 1.2X mirror the relative phenotypic change 1.2X observed in Figure 4E (55%/45%).

Further, as suggested by the reviewer we have performed GSEA to compare sepsis induced changes with effector vs memory CD8 T cells to further highlight the compositional shift from T_EM_ to T_CM_. Indeed, P14s from Sham hosts were biased toward effector cells while P14s from CLP hosts were biased toward memory cells. This reflects the time-dependent shift from effector to central memory, mirrored in the sepsis-induced changes in memory CD8 T cells (Figure 6—figure supplement 1and line 355-361).

Whether if this phenomena is common with endogen cells, and if this is associated with alteration of cell function is unknown.

While we did not perform RNA-seq/ATAC-seq on total endogenous cells for the reasons outlined in response to point number 1 we do believe the virus-specific memory P14 CD8 T cells are a faithful sensor population as they have behaved similarly when analyzed in comparison to the endogenous CD8 T cells (Martin et al., 2015 Plos Pathog. e1005219; Khan et al., 2019 Nature 571(7764):211-218). Further, in the event of differences it would be impossible to rule out that the differences were not due to changes in the pool of activated cells (inclusion of sepsis responsive effector or memory CD8 T cells or cross-reactivity epitopes between the LCMV infection with the septic event) rather than true differences between memory P14s and endogenous CD8 T cells.

Notably, is the production of IL-2 during sepsis similar in CD62L+ and CD62L- cells?

CD62L is shed from the cells surface during antigen encounter (Herndler-Brandstetter et al., 2005 J Immunol. 175(3)1566-1574) it is difficult to determine whether the cells CD62L- cells from the CLP hosts are actually generating IL-2 or if it is merely (likely) that the CD62L+ cells shed their CD62L following antigen encounter. However, given that it is widely reported that CD62L+ cells preferentially produce IL-2 (Martin and Badovinac 2018 Front. Immunol. 2692; Wherry et al., 2003 Nat. Immunol. 4:225-234) and that the increase in IL-2 production coincides with our similar observation of an increase in CD62L we feel that it is more reasonably the CD62L+ cells that are contributing the IL-2 following stimulation in Figure 7.

Overall, the RNAsequencing analyses is difficult to interpret due to the highly curated presentation and lack of fuller datasets.

We hope that the addition of the gene tables and the discussion mentioned provide now enough clarity to reviewers (and hopefully other scientist that will read our manuscript) to interpret/understand these data sets.

Reviewer #1:In this work, the authors endeavoured to understand how CD8 T cells, and preexisting memory CD8 T cells, might be influenced by sepsis. This is a very important question. They uncover interestingly that memory cells can still respond to subsequent infections, but are indelibly modulated by their exposure to the septic environment. This work provides interesting insights to the modulation of T lymphocytes and subsequent responsiveness post sepsis.

We appreciate the reviewer’s assessment our manuscript.

This manuscript addresses the important issue of how sepsis influences the function particularly of CD8 T cells. It shows that while antigen-specific T cells respond to infection following transient sepsis and paralysis, embedded changes occur in the T cells influencing their long-term performance. Although the changes in T cells are interesting, these effects are not dissociated from the extrinsic influences of other cell types that might also be affected by sepsis.

We recognize that effects in memory CD8 T cell subsets are not the sole influence by which sepsis alter host immune status. While this study is focused on pre-existing memory CD8 T cells future assessments should/would harmonize these findings with alterations in other cell types.

Page 4, line 91/92: There appears to be a word or words missing.More recent references eg. Roquilly et al., Immunity and Nature Immunology have not been referenced.In the human samples, a gap in the study is that admission may not correspond to the onset of sepsis. Delineation of a surrogate for the timing would be useful and some de-identified data to provide a temporal perspective of the onset of sepsis. One possibility may be the timing of lymphopenia. Part of the information is provided in Table 1.The initial data is developed through characterisation of the proportions and numbers of different subsets of CD8 T cells. The hypotheses that the differential expansion/contraction of these subsets is based on our knowledge of normal responses. It is not entirely clear that this will be the case during sepsis. Using clustering based on known surface receptor expression, the authors designate 8 different subsets which they posit a pattern of temporal development. They then go on to endeavour to uncover a mechanism for the altered activity of the CD8 T cells and use bulk RNAseq. In this analysis they identify 3 clusters. It is not clear exactly how the 8 subsets identified by flow cytometry and the sequence clusters relate to each other. Cluster 3 is composed of 57 genes that are changed but none of the genes in this cluster were mentioned nor provided in a table for reviewer consideration. The RNAsequencing analyses is difficult to interpret due to the highly curated presentation and lack of fuller datasets.

These comments have been addressed in the above response to the editor.

Overall, this is an interesting study but the data are presented in quite a superficial manner limiting the impact of the work.

We appreciate that these represent some of the more initial aspects of the long-term impacts of sepsis on pre-existing memory CD8 T cells but contend that they provide a substantive advance, uncover interesting and previously unknown biology, and lay the groundwork for future experiments that will further analyze/uncover/define role of sepsis in shaping vaccine and/or infection induced memory CD8 T cell pool and ability of the host to respond to pathogen (re)-encounter.

Reviewer #2:Isaac J. Jensen et al., investigated in septic humans and in a mice of peritonitis the time course of the modifications of memory T cells during and after sepsis. They tracked in vivo the fate of antigen-specific memory T cells by using an elegant Antigen-specific T cells transfer whose proliferation is induced by a first viral infection. Once the virus-specific memory T cells are well settled, they induced an intraabdominal sepsis to investigate the modification of memory T cells during and after sepsis. They found in septic humans that while the percentages of CD8 T cells among lymphocytes remained unchanged, the rates of proliferating naïve and memory CD8 T cells were increased during sepsis. In the mice models, they observed that the survival of memory T cells decreased during sepsis, but their number rapidly returned to control values due to high rate of in vivo proliferation. Yet, the recovery of the number of memory T cells is associated with modifications of the proportions of effector vs. central memory T cells, and of the tissue localization. Transcriptomic and epigenetic analyses confirmed modifications of the CD8 T cells functional programming during sepsis, with upregulation of cell survival and proliferation functions. Finally, the authors demonstrated that the cytokine production of antigen-specific memory T cells is not decreased during sepsis, and IL-2 being even increased. in vivo, the gain of function of antigen-specific CD8 T cells (high proliferation, high IL-2 production) are not associated with higher control of viral load during reinfection. Altogether, the authors demonstrated that while memory CD8 T cells gain function during sepsis, it is associated with a poorer control of viral infection. These data add in an interesting way to the ongoing discussion on whether sepsis induced training of immunity (gained functions and increased resistance to infection) or tolerance / immunosuppression (loss of functions and increased susceptibility to infection).

We appreciate the reviewer’s assessment of our manuscript.

The conclusions of this paper are mostly well supported by data, but some aspects of data analysis need to be clarified and extended.1) the mice model used to discriminate endogen and transferred memory T cells is underexploited. While P14 T cells are LCMV-specific, they are not specific to the antigens produced during CLP surgery. So the reasons to discriminate endogen (Thy 1.1neg) vs. P14 (Thy 1.1pos) during CLP are not clear. Most of the time, endogen and P14 CD8 memory T cells have the same response (Figure 2C, 4D-E) while this information is missing and only P14 response is described in most of the experiments (Figure 3, Figure 4B-C, 4G, 5, 6 and 7). The comparison of endogen CD8 memory T cells with P14 should be consistent throughout the study since the role of TCR signaling could be of importance (as suggested by the increased Cish gene which is involved in TCR functional inhibition, see RNAseq, Figure 5H). Finally, the in vivo functional assay of the memory T cells response (Figure 7E-G) does not exclude a bystander role of endogen cells since IL-2 production can act on both cell types.2) The description by flow cytometry of different subsets of P14 cells during sepsis (Figure 3C-H) is of interest but the mechanisms explaining the increased in cluster 8 (CD62L+) is not clear. Are the modifications in phenotype observed in Figure 3 explained by the transcriptomic activity of cells? The finding of a CM and EM signature in the transcriptomic analyses would have strengthened the results. Whether if this phenomena is common with endogen cells, and if this is associated with alteration of cell function is unknown. Notably, is the production of IL-2 during sepsis similar in CD62L+ and CD62L- cells?

These comments have been addressed in the above response to the editor.

As a summary, data are sounds, but the demonstration that the alterations are specific, or not, to any memory CD8 T cells subsets; and are antigen-specific or not, would have significantly increased the gain of knowledge.

We recognize the reviewers point and this would be a highly relevant aspect for future consideration. While it would be ideal to observe different populations behaving discretely this is technically challenging, particularly so if the distinctions rely on cross-reactivity between existing memory CD8 T cells and gut microflora released during the septic insult. Text addressing these points has been incorporated in the discussion (line 456-460).

In general, the paper is difficult to follow because the studied cells frequently between Figures: endo. vs P14, CD62L+ vs. CD62Lneg, then 14 alone.1 – I would recommend to analyse endogen and P14 cells together throughout the manuscript. Indeed, I think that more than the endogen vs. transfer feature, P14 cells are likely not responding directly to CLP-derived antigens.

We agree with the reviewer’s assessment in principal and have in the above response attempted to explain our rationale for lacking the analysis of the total endogenous memory CD8 T cell population in some cases and providing the additional results with bona-fide endogenous memory CD8 T cells where possible (new Figure 7—figure supplement 2 and line 384-386).

2 – The definition of cell survival (Figure 2C) is not clear to me and should be better explained.

This comment has been addressed in the essential revisions portion (please see above).

3 – While the data of CD62L+ vs CD62L- subsets are of interest (Figure 3), this information is not exploited in the RNAseq and ATAC-seq analyses. The comparison with public data set of effector memory and central memory T cells would likely reinforced the message of differential composition of these subsets.

The reviewer’s point is of interest. We performed the requested analysis, discussed response to the above editor comment. Data are presented in Figure 6—figure supplement 1 and discussed in the text (line 355-361).